# Assessing soil and land health across two landscapes in eastern Rwanda to inform restoration activities

Leigh Ann Winowiecki[1], Aida Bargués-Tobella[2], Athanase Mukuralinda[3], Providence Mujawamaria[3], Elisée Bahati Ntawuhiganayo[3], Alex Mugayi[4], Susan Chomba[5], Tor-Gunnar Vågen[1]

[1]World Agroforestry (ICRAF), Nairobi, Kenya
[2] Department of Forest Ecology and Management, Swedish University of Agricultural Sciences (SLU), Umeå, Sweden
[3] World Agroforestry (ICRAF), Kigali, Rwanda
[4] World Vision- Rwanda, Kigali, Rwanda
[5] World Resources Institute, Nairobi, Kenya
*Correspondence to*: Leigh Ann Winowiecki (L.A.Winowiecki@cgiar.org)

## Abstract

Land degradation negatively impacts water, food and nutrition security and is leading to increased competition for resources. While landscape restoration has the potential to restore ecosystem function, understanding the drivers of degradation is critical for prioritizing and tracking interventions. We sampled 300-1000m$^2$ plots using the Land Degradation Surveillance Framework across Nyagatare and Kayonza districts in Rwanda to assess key soil and land health indicators, including soil organic carbon (SOC), erosion prevalence, vegetation structure and infiltration capacity, and their interactions. SOC content decreased with increasing sand content across both sites and sampling depths and was lowest in croplands and grasslands, compared to shrublands and woodlands. Stable carbon isotope values ($\delta^{13}$C ) ranged from -15.35 to  -21.34 ‰ indicating a wide range of historic and current plant communities with both C3 and C4 photosynthetic pathways. Field-saturated hydraulic conductivity ($K_{fs}$) was modeled, with a median of 76 mm h$^{-1}$ in Kayonza and  62 mm h$^{-1}$ in Nyagatare, respectively. Topsoil OC had a positive effect on Kfs, whereas pH, sand and erosion had negative effects. Soil erosion was highest in plots classified as woodland and shrubland. Maps of soil erosion and SOC at 30m resolution were produced with high accuracy and showed strong variability across the study landscapes. These data demonstrate the importance of assessing multiple biophysical properties in order to assess land degradation, including the spatial patterns of soil and land health indicators across the landscape. By understanding the dynamics of land degradation and interactions between biophysical indicators, we can better prioritize interventions that result in multiple benefits, as well as assess the impacts of restoration options.

## 1. Introduction

Land degradation is inextricably linked to livelihoods and negatively impacts over 3.2 billion people each year globally (IPBES, 2018). Land degradation also adversely affects the resilience of social-
ecological systems to climate change by reducing their adaptive capacity. Therefore, the combined impacts of land degradation and climate change represent a significant risk to global food security (Webb et al., 2017), particularly when considering positive feedback effects between processes such as more erratic and intense rainfall events and soil erosion. Similarly, land degradation strongly impacts the loss of biodiversity globally, further reducing the adaptive capacity of ecosystems in the face of
climate change (Gisladottir and Stocking, 2005), which means that we cannot tackle any of these global challenges in isolation.

Efforts to avoid, reduce and reverse land degradation are therefore critical if the Sustainable Development Goals (SDGs) are to be achieved (IPBES, 2018).  SDG 15.3, Life of Land, has set
ambitious targets for land degradation neutrality (LDN), combining belowground indicators, i.e., soil organic carbon (SOC), and aboveground measures (net primary productivity and land use) (Cowie et al., 2018). In line with this thinking, forest and landscape restoration aims to regain ecological functions, including biodiversity and soil function, and enhance human well-being across landscapes (Chazdon, 2008; Chazdon et al., 2016). The UN Decade on Ecosystem Restoration (2021-2030) offers promising
opportunities to bring together the global community to scale efforts across the globe. These efforts highlight the complexity of ecosystems and that multiple biophysical and socio-economic factors need to be considered when targeting, planning, implementing and tracking restoration on the ground. This includes understanding the spatial and biogeochemical variations of the soil ecosystem, which is the foundation for biophysical land restoration efforts, given its role in global net primary productivity.
The global community acknowledges the need for long-term monitoring networks across diverse environments (Navarro et al., 2017; Sachs et al., 2010), including those focused on soil monitoring (Guerra et al., 2021; Lehmann et al., 2020; Vermeulen et al., 2019), in order to better understand drivers and interactions as well as track progress of interventions.  However, many assessments of land
degradation and restoration suffer from (i) disagreements about the definition of land degradation, (ii) a conundrum of indicators that are often not feasible to measure and hence operationalize, and (iii) a lack of rigorous science-based analytical frameworks (Vågen, 2015). Indicators are critical when assessing ecosystem health and tracking progress toward restoration targets or climate actions and can be important communication tools for decision-makers. Indicators should be readily measurable,
quantifiable and encompass the complexity of various drivers.

The call for soil degradation and resilience indicators is not new (Lal, 1997); however, scientific research around the concept of soil health continues (Lehmann et al., 2020). We argue that a coherent set of indicators collected using consistent measurement methods is needed to address the completely of
ecosystem function. SOC is widely accepted as a key indicator of soil health due to its influence on multiple indicators and its response to aboveground processes, including land management (Deb et al., 2015; Paustian et al., 2019; Shikuku et al., 2017). In addition, SOC is seen as a key indicator to monitor progress on a number of SDGs (Lorenz et al., 2019). Soil erosion is arguably the most important indicator of land degradation and also one of the most widespread forms of degradation worldwide

(Bennett, 1939; Lal, 2003; Pimentel, 2006; Vågen and Winowiecki, 2019). In addition exchangeable base cations provide a measure of available nutrients and soil pH provides a measurement of potential constraints such as acidity. Land cover and vegetation structure play a key role in terms of driving soil organic carbon dynamics in landscapes while also influencing land degradation processes such as soil erosion. Therefore, indicators such as tree density within various vegetation structure classes and overall

tree diversity provide useful information for informing restoration interventions around reforestation (Di Sacco et al., 2020). The use of carbon isotopes provides further insights on vegetation shifts as $\delta^{13}C$ values in the soil reflect the photosynthetic pathway of the aboveground vegetation (Boutton et al. 1998). Soil infiltration capacity is another well-established indicator of soil health, in particular of the soil's physical status and its hydrological functioning (Allen et al., 2011). Soil infiltration capacity

influences the recharge of soil and groundwater stores and the generation of surface runoff, with implications for erosion and flooding occurrence (Hillel, 1998).

Given the heterogeneity of landscapes, spatial information on the distribution of these indicators needs to be made at relevant spatial scales (i.e., at the farm, landscape, and regional levels). Furthermore,

interactions between these indicators need to be considered explicitly. Recent advancements in spatially explicit assessment of soil and land health that combine field-based campaigns with data analytics and earth observation are now paving the way for improved methods of biophysical characterization of multiple indicators (Vågen et al., 2016) while providing an opportunity to enable science-based monitoring approaches that can be applied in restoration prioritization (Winowiecki et al., 2018) as well

as for communication with decision-makers (Vågen et al., 2018a).

In Rwanda, land degradation continues to be a critical challenge. To combat this, Rwanda set a goal to achieve land degradation neutrality by 2030 and, in 2011, Rwanda was the first country in Africa to commit to a restoration target of degraded lands and forests under the Bonn Challenge, pledging to

restore 2 million ha, corresponding to 76% of the country. Underlying causes of land degradation in the country include unsustainable farming and grazing practices, overexploitation of forests and woodlands, settlements and urbanization (Bizimana, 2018). One of the major processes of land degradation in Rwanda is accelerated soil erosion, which is driven by unsustainable agricultural practices, particularly in steeply sloping lands (Karamage et al., 2016). This is further exacerbated by intense rainfall events,

resulting in increased rainfall erosivity (Rutebuka et al., 2020) and the increasing energy demands of a growing population resulting in deforestation and loss of vegetation cover in general (Mukuralinda et al., 2016). Soil erosion is severe with mean national rates of 250 Mg ha$^{-1}$ yr$^{-1}$, and studies showing as much as 421 Mg ha$^{-1}$ yr$^{-1}$ in croplands (Karamage et al., 2016).

Considering that the agricultural sector contributes significantly to the national economy and that 90% of the population depends on agriculture for their livelihoods, tackling land degradation and restoring degraded land is of critical importance for Rwanda. Studies suggest that investments in soil conservation and land productivity are contributing to reduced land degradation and increased agricultural productivity in Rwanda (Bidogeza et al., 2015; Byiringiro and Reardon, 1996; Fleskens,

2007; Bizoza and De Graaff, 2012; Karamage et al., 2016). For example, various forms of terracing have been implemented across Rwanda to specifically curb the negative effects of intensive farming on

steep slopes on soil fertility and soil loss (Kagabo et al., 2013). Studies also show that terracing coupled with building organic matter has the potential to be financially profitable when access to labour and manure is facilitated (Bizoza and de Graaff, 2012). Furthermore, there is a real need for a systems

approach to sustainable agricultural intensification that spans from appropriate technologies to institutional and policy-level support (Schut et al., 2016; Vanlauwe et al., 2014). In addition, agroforestry approaches have also been suggested to meet the multiple demands of farming households, including in Rwanda (Liyama et al., 2018). However, improved targeting of interventions and tracking of progress overtime could both improve not only the success of restoration efforts but demonstrate

which options works best under the various conditions.

In this study, we applied a systematic approach to collecting data on soil health and land degradation indicators, including the use of soil spectroscopy, using the Land Degradation Surveillance Framework (LDSF) (Vågen and Winowiecki, 2020) across agricultural-dominated landscapes in eastern Rwanda.

Studied indicators included SOC, erosion prevalence, vegetation structure, tree density and species diversity, topsoil field-saturated hydraulic conductivity (a proxy for steady-state infiltration capacity), soil texture, pH and exchangeable bases. Specific objectives of this study were to: 1) Assess soil and land health indicators across two landscapes; 2) Identify biophysical constraints; 3) Develop maps of soil erosion hotspots and variations in SOC for restoration interventions, based on the hypothesis that

remote sensing (spectral) data can be used to predict erosion and SOC. We also assessed the relationship between inherent soil properties, such as texture, and SOC, the hypothesis being that factors such as sand content create constraint envelopes in terms of variations in SOC. Another hypothesis addressed in the study was related to whether there is a positive effect of SOC on field-saturated hydraulic conductivity when we consider data from across diverse landscapes. We also assesses the

influence of other soil properties on field-saturated hydraulic conductivity, in addition to human-induced processes such as soil erosion. Finally, we assessed the current status of vegetation structure across the landscape, in addition to tree density and tree species diversity, and conducted spatially-explicit assessments of SOC for eastern Rwanda.

## 2. Methods

### 2.1. Site Description


The LDSF was implemented in two districts in eastern Rwanda, Nyagatare and Kayonza. Nyagatare is the largest dairy district in Rwanda and is characterised by two main seasons: one long dry season and a short rainy season. Its annual average temperature varies between 25.3 and 27.7 °C, and it receives an annual rainfall of 827 mm. However, rainfall patterns have become increasingly unpredictable and

variable. The average altitude is 1,513 m. It consists of gently sloping hills separated by low granitic valleys. The vegetation type was originally savannah vegetation and some gallery forests. From 2009 to 2019, there was a net loss of forest cover with deforestation and afforestation rates at 34% and 18%, respectively (MoE, 2019). The major economic activity is subsistence farming, while the main source of cooking energy is fuelwood. Multiple crops are cultivated in Nyagatare including, maize, beans,

groundnut, cassava, irish potatoes, banana, and yams, among others. Some areas have been cultivated for 100 years, but the majority of the agricultural expansion in the district took place between 1973 and 1995. The dominant soil types in Nyagatare are Ferralsols (Oxisols) with shallow Leptosols on hillsides, according to data from the Ministry of Agriculture (MINAGRI).

Kayonza district has a mean altitude of 1,428 m and a mean annual rainfall of 919 mm (NISR, 2012). It is prone to long drought events with two principal seasons, a long dry period and a short rainy season. Crops cultivated in Kayonza include beans, banana, cassava, maize, irish potato, sorghum, cocoa yams, among others. Most of the area has been cultivated for over 50 years, with mining activities also taking
place. Dominant soil types in the Kayonza site are Ferralsols and Leptosols, with Histosols in lower-lying areas.

## 2.2. Field Sampling using the Land Degradation Surveillance Framework

The LDSF is a systematic methodology to conduct landscape-level assessments of soil and land health
based on a consistent set of indicators and field protocols. The framework was developed by the World Agroforestry (ICRAF) in response to the need for a consistent field method and indicator framework to assess soil and land health at the landscape scale. The LDSF has been applied in several projects across the global tropics (Vågen et al., 2016; Vågen and Winowiecki, 2020, 2019) and is currently one of the largest ecosystem health databases globally, with data from more than 30,000 plots in over 40 countries.
The LDSF uses a hierarchical sampling design to simultaneously measure and assess several land and soil health indicators, including vegetation cover and structure, current and historical land use, erosion prevalence, soil infiltration capacity, soil texture, soil pH and SOC. An LDSF site is a 100 km$^2$ area stratified into 16-1 km$^2$ clusters, each containing 10-1000 m$^2$ plots and 4-100 m$^2$ subplots (L. Winowiecki et al., 2016). The hierarchical sampling design enables robust analysis of drivers of
degradation as well as the production of predictive maps of soil health indicators, for example, SOC (Vågen et al., 2018b). The two LDSF sites in this study were randomized within each of the districts. The field team navigated to the randomized plots and set up the four circular subplots within the plot.

Measurements took place at the plot and subplot levels. All plots were geo-referenced to better than 5m
accuracy. Vegetation structure was classified at the plot level using the FAO Land Cover Classification System (LCCS), which was developed in the context of the FAO-AFRICOVER project (Di Gregorio, A., and Jansen, 2000). Specifically, plots were classified as either annual cropland, grassland, shrubland, woodland, or forest vegetation structure. In the LDSF, trees are classified as woody vegetation above 3 m tall, whereas woody plants 1.5-3 m in height are classified as shrubs. All trees
were counted and identified to species level in each of the four subplots per plot. Soil erosion was scored and classified in each subplot (n=4) per plot. Specifically, each subplot was visibly assessed for erosion (i.e., rill, sheet or gully), otherwise the plot was marked as having no erosion. Erosion scores (presence (1) or absence (0)) were used in the statistical analysis. Soil samples were collected using a soil auger at the center of each subplot at two depths (0-20 cm (topsoil) and 20-50 cm (subsoil)). Soil

samples were combined from the four subplots into one composite sample per LDSF plot and depth increment.

Infiltration capacity was measured at three plots per cluster in each site using single ring infiltrometers (Bouwer, 1986) to assess variation across land uses and soil types. Soil infiltration capacity into dry soils follows a predictable temporal pattern: it is high in the early stages of infiltration and tends to decline gradually as the soil moisture content increases until it eventually approaches a nearly constant rate known as steady-state infiltration capacity (Horton, 1940). This steady-state rate is independent of the initial soil water content and approximates the soil's saturated hydraulic conductivity. Infiltration measurements were carried out at the center of each plot using a metal cylinder with an inner diameter of 15.6 cm and 20 cm in height for two hours and a half to ensure capturing steady-state conditions.

Field-saturated hydraulic conductivity (Kfs) (Reynolds and Elrick, 1990) was calculated from the infiltration data using the analytical formula proposed by Nimmo et al. (2009). First, infiltration rates were corrected for non-constant falling head and subsurface lateral spreading effects. For each plot, an asymptotic function was then fitted to its corrected infiltration curve using the *nls.multstart* package in R (Padfield and Matheson, 2018) to obtain the asymptote, which represents Kfs.

The effects of soil and land use and land cover variables on Kfs were assessed with linear mixed effects models using the *lme4* package (Bates et al., 2015) in R. Random effects intercept models were fitted using the lmer function, with a random intercept for each level of site and for each level of cluster within site (nested grouping factors). To assess statistical significance of fixed effects, we used the *lmerTest* package in R (Kuznetsova et al., 2017).

The rationale behind the use of the LDSF in the current study was that it has been applied across a wide range of landscapes in the global tropics and has been shown to be robust in terms of assessing soil and land health in landscapes. It uses a standardized set of indicators that are consistently sampled and quantified, allowing for comparative studies between sites or landscapes. Also, the LDSF has been successfully applied in other studies for the mapping of indicators of soil and land health when used in combination with remote sensing satellite data (Vågen and Winowiecki, 2019, Vågen et al., 2013b).

### 2.3. Laboratory Methods

Upon collection, all soil samples were processed locally in Rwanda, air-dried and ground to pass through a 2-mm sieve. Air-dried and ground samples were packed and shipped to the ICRAF Soil-Plant Spectral Diagnostics Laboratory in Nairobi, Kenya. Further grinding was then conducted on a subsample using a Retsch motor grinder to attain a particle size between 20 and 53 microns. This subsample was analyzed in triplicate for MIR absorbance using a Tensor 27 HTS-XT from Bruker Optics in the ICRAF Soil-Plant Spectral Diagnostics Laboratory in Nairobi, Kenya. The measured wavebands ranged from 4000 to 601 cm$^{-1}$ with a resolution of 4 cm$^{-1}$. Processing of the MIR spectra included computing the first derivatives using a Savitsky-Golay polynomial smoothing filter

implemented in the locpoly function of the *KernSmooth* R package (Wand, 2015) as outlined in Terhoeven-urselmans et al. (2010).

Wet chemistry reference analysis was conducted on 10% of the collected soil samples (n=32 samples per site, 16 topsoil and 16 subsoil samples). Soil pH and exchangeable bases were measured at Crop
Nutrition Laboratory Services in Nairobi, Kenya. Soil pH was analyzed in a 1:2 $H_2O$ mixture that was shaken for 30 min at moderate speed on a horizontal shaker then let stand for 20 min before reading on a Eutech Cyberscan 1100 pH meter. Exchangeable bases were extracted using the Mehlich-3 method after five minutes on a reciprocating shaker. The filtrate was analyzed for base cations: potassium (K), calcium (Ca), magnesium (Mg) and sodium (Na) using an ICP OES (Model-Thermo iCAP6000 Series).
Total nitrogen, organic carbon and stable carbon isotopes ($\delta^{13}C$) were measured by dry combustion using an Elemental Analyzer Isotope Ratio Mass Spectrometry (EA-IRMS) from Europa Scientific after removing inorganic C with 0.1 N HCl, at the IsoAnalytical Laboratory located in the United Kingdom. Stable carbon isotopes were expressed as $\delta^{13}C$ in parts per mile (‰) relative to the V-PDB (Pee Dee Belemnite) standard. Sand content was measured using a Laser Diffraction Particle Size Analyzer
(LDPSA) from HORIBA (LA 950) after shaking each soil sample for four minutes in a 1% sodium hexametaphosphate (calgon) solution at the World Agroforestry Centre (ICRAF) Soil-Plant Spectral Diagnostics Laboratory in Nairobi, Kenya.

**2.4. Prediction of soil properties from MIR soil spectroscopy**

Soil samples with both MIR spectra and associated wet chemistry data were used to train (calibrate) predictive models in order to simultaneously predict multiple soil properties using random forest (RF) regression models (Vågen et al., 2016). In the RF algorithm, many decision trees are built, each on a bootstrap sample, based on a random subset of the input MIR spectra and these trees are combined to
predict the different soil properties. The total number of reference samples used for model development and testing were 10,820 for SOC, 7,305 for soil pH, 4,322 for soil texture and 1,657 for $\delta^{13}C$. In training the prediction models, we randomly selected 70% of the samples for each soil property, keeping the remaining 30% out for testing of the models. We then calculated $R^2$ and Root Mean Square Error of Prediction (RMSEP) values for the training and test dataset to assess model performance.

**2.5. Landscape-level mapping of soil erosion and SOC**

We used LDSF soil and field data from a total of 30,853 sites in 40 countries, including the two sites from this study, to generate prediction models and map SOC and soil erosion based on Landsat 8 reflectance data. The approach we followed in this study is described in Vågen et al. (2013), but we applied Landsat 8 rather than Landsat 7 and a larger database of LDSF sites. A Landsat 8 spectral
library was built for all of the LDSF plots by extracting surface reflectance values for each band, matching remote sensing data acquisition to within six months of field survey dates. Cloud masking was conducted prior to surface reflectance extraction. We then used the annual median reflectance values for

each band as input into the prediction models for SOC and erosion in order to map SOC concentrations (gC kg$^{-1}$) and the probability of erosion (in %) for each 30m Landsat pixel.


We assessed the performance of the prediction models in a similar manner as for the soil MIR predictions by using 70% of the plots to train the models and the remaining 30% to test performance. For erosion, we assessed model performance by calculating the percentage of correctly classified test instances relative to observed instances, expressed as a confusion matrix, and by calculating the

Receiver Operating Characteristic curve (ROC) (Bradley, 1997), which evaluates the accuracy of a model by considering errors that are either *false positives* or *false negatives*.

## 3. Results

### 3.1. Vegetation structure and diversity in the LDSF plots sampled

LDSF field surveys took place between October and November 2018. In total, 151 plots were sampled

in Kayonza and 149 plots were sampled in Nyagatare. Both sites were dominated by annual cropping systems, with 68% of the sampled plots in Kayonza classified as cultivated and 89% in Nyagatare. Other vegetation structure classes included shrubland (19% in Kayonza, 3.4% in Nyagatare), woodland (9.3% in Kayonza and 7.4% in Nyagatare) and grassland (3.3% in Kayonza). Mean tree density was higher in Nyagatare (120 tree ha$^{-1}$) compared to Kayonza (68 tree ha$^{-1}$). Overall, this level of tree

density is low, and the higher tree densities only occurred in woodlots of *Eucalyptus spp.* (Figure 1). Mean tree density in croplands was 57 tree ha$^{-1}$ in Kayonza and 35 tree ha$^{-1}$ in Nyagatare. The plots with higher tree density in croplands were dominated by *Eucalyptus spp*. In total, 62 unique tree species were identified in the two LDSF sites. The most common species was *Eucalyptus spp.*, followed by *Grevillea robusta*, *Euphorbia tirucalli*, *Ricinus communis*, *Mangifera indica*, *Carica papaya* and *Senna*

*spectabillis* (Figure 2). Differences were observed between the two LDSF sites, most notably that *Jatropha curcas* was only found in Kayonza and *Senna singueana* was only found in Nyagatare. (Figure 2). In summary, 48 unique tree species were observed in Kayonza and 39 species in Nyagatare. This level of tree diversity is considered quite low, with a low occurrence of most species, low occurrence of only a few indigenous species, and dominance of *Eucalyptus spp*. For example, 171 (56%) of the

sampled plots had *Eucalyptus spp.*, including 125 of the cropland plots (53%).

### 3.2. MIR prediction results for soil properties

Prediction performance was good for the soil properties included in the study, including for the prediction of δ$^{13}$C, as summarised in Table 1. The prediction model performance for δ$^{13}$C is similar to

that reported by (Winowiecki et al., 2017) when predicting δ$^{13}$C based on near-infrared (NIR) spectroscopy. Figure 3 shows predicted versus measured SOC and δ$^{13}$C, respectively, for Nyagatare and Kayonza, showing good model performance across a wide range of SOC and δ$^{13}$C values, respectively.

### 3.3. Soil properties and erosion prevalence

Soil properties for top- and sub-soil samples for Kayonza (n= 151, 136) and Nyagatare (n= 149, 145) LDSF sites are presented in Table 2. Density plots for the soil variables demonstrate the variability between and within the sites (Figure 4). Overall, pH values were low across the two sites, with statistical differences in topsoil pH values between sites (P<0.001); mean topsoil pH was 5.65 in Kayonza and 5.89 in Nyagatare. This level of pH can potentially limit agricultural production. Both sites had low overall exchangeable bases (Ca, K, Mg, Na), as 8 cmol$_c$ kg$^{-1}$ is considered critically low for agricultural productivity. Kayonza had significantly higher clay content and lower sand content compared to Nyagatare (P<0.001). Kayonza had statistically higher topsoil OC content (20.9 g kg$^{-1}$) compared to Nyagatare (17.3 g kg$^{-1}$) (P<0.001). Figure 5 shows the relationship between sand content and SOC content, with SOC increasing with decreased sand content for both sites and depth intervals. This demonstrates the important control of inherent soil properties, i.e., sand content, on SOC. The same pattern was observed in each vegetation structure class. However, SOC was lowest in the cropland and grassland plots compared to shrublands and woodlands (P<0.001). Average δ$^{13}$C was 18.9 ‰ in Kayonza and -19.2 ‰ in Nyagatare, which indicates that these are mixed C3-C4 systems. We also assessed the variation of stable carbon isotopes within and between the vegetation structure classes (Figure 6). While there were some distinctions between classes, namely more negative isotope values in woodlands compared to croplands, overall δ$^{13}$C values were relatively similar. The observed overlap is likely due to the high occurrence of *Eucalyptus spp.* (even in cropland plots) and the fact that woodland plots were previously cultivated, resulting in the mixed C3-C4 signal.

Kayonza had a higher soil erosion prevalence, with 45% of the plots considered severely eroded, compared to 27% of the sampled plots in Nyagatare. The dominant erosion categories were rill and sheet. Severe erosion was more prevalent in woodland (91%), shrubland and grassland (77%) compared to cropland (25%). This is most likely given the high prevalence of terracing in the region as well as the location of the cropping fields compared to woodland and bushland. For example, the average slope for the plots classified as cultivated was seven degrees compared to 19 degrees for the other vegetation structure classes. There was no statistical difference in SOC in severely eroded and non-severely eroded plots; however, cropland plots were the dominant category across the landscape and only 24% of cropland plots were classified as severely eroded.

### 3.4. Saturated hydraulic conductivity

Median topsoil field-saturated hydraulic conductivity (Kfs) in Kayonza was 76 mm h$^{-1}$, whereas in Nyagatare it was 62 mm h$^{-1}$ (Figure 7). In Kayonza, Kfs was not only higher but also more variable than in Nyagatare, with an interquartile range (upper quartile – lower quartile) of 77 mm h$^{-1}$ and 42 mm h$^{-1}$, respectively.

Results from the linear mixed effects (lme) models showed that the presence of erosion and pH had both a significant negative effect on Kfs (P < 0.025 and P < 0.016, respectively). Topsoil OC had a nearly significant (P <0.082) positive effect on Kfs, whereas sand content had a significant negative effect (P

<0.033).We could not assess the effect of vegetation structure on Kfs, as most of the plots where infiltration was measured were on cropland.

## 3.5. Soil mapping

Soil erosion prevalence was predicted with a high degree of accuracy using Landsat 8 satellite data, with an out-of-bag prediction (OOB) error of 14%. The OOB prediction error-rate is based on a bootstrap sample of about 37% of unused test observations and represents a robust assessment of accuracy. Further to the calculation of the OOB error-rate, the receiver operator characteristics (ROC) curve also indicates good model performance with the area under the ROC curve (AUC) calculated at 0.86. These results are consistent with previous studies using remote sensing to predict erosion (Vågen et al., 2013; Vågen and Winowiecki, 2019). Given the level of accuracy, we applied the random forest model to Landsat 8 imagery for 2018, generating a map of soil erosion at 30-m resolution for the study area. Hot spots of erosion are shown in red and yellow in the map in Figure 8, representing areas where erosion prevalence is predicted to be over 75% in 2018, some areas also having extreme erosion (>75%). As we can see from this map, there is high spatial variability of erosion across eastern Rwanda.

The prediction model performance for SOC was also good, with an $R^2$ of 0.82 based on the OOB prediction results from the random forest model and testing of the prediction model on an independent test dataset (Figure 9). The map of SOC (Figure 10) shows high levels of variation in SOC across the study area with particularly low SOC in Nyagatare district, except for wetlands along rivers and in forested areas in the west of the district. Similarly, in Kayonza ditrict, the map shows higher SOC in protected areas and in lower lying areas, including in wetlands in the eastern part of the district.

## 4. Discussion

The LDSF was used to assess soil and land health indicators across two landscapes in eastern Rwanda. Both sites (Kayonza and Nyagatare) were dominated by annual cropping systems, and both sites had overall low tree densities and low tree diversity. *Eucaltypus spp* dominated both the woodland and cropland systems in both sites, followed by *Grevillea robusta*. *Jatropha curcas* was observed only in Kayonza and *Senna singueana* was only observed in Nyagatare. These data have important implications for restoration activities. For example, tree planting is in the global spotlight as a restoration activity with high potential for climate change mitigation, while providing multiple other ecosystem services (Bastin et al., 2019). However, the global community acknowledges that tree planting and reforestation must do down taking into account multiple environmental and socio-economic considerations. For example, prioritize appropriate areas to restore, use natural regeneration, maximize biodiversity, among other principles (Di Sacco et al., 2021). In Rwanda, there are multiple tree planting campaigns funded by the government as well as within the development sector. These data demonstrate a real opportunity to improve tree biodiversity across the landscape, including on cultivated fields. While woodlands reportedly had higher SOC content compared to the other vegetation structure classes, woodlands also

had mixed land use history, from native vegetation to being cultivated, leading to the high variation in SOC values. Our findings of low tree species diversity are similar to those of other studies from other regions of Rwanda (Bucagu et al., 2013; Liyama et al., 2018), highlighting the opportunity for the strategic inclusion of useful and appropriate species that fulfill multiple ecosystem benefits, including the inclusion of indigenous tree species on farms.


This paper highlights the importance of assessing key soil and land health indicators, most notably SOC and soil erosion. The concept of soil health goes beyond individual indicators and is more about building and maintaining a functioning soil ecosystem to provide and support multiple ecosystem services and functions. Lehman et al. (2020) discussed the shift of focus of soil assessments from crop

productivity to human health, climate change adaptation and mitigation and water quality and quantity. This shift acknowledges the linkages across multiple indicators, and this information can be used to prioritize interventions to maximize benefits and minimize tradeoffs.

For example, inherent soil properties, such as soil texture, are influenced by parent material, yet, they

can impact dynamic soil properties. For instance, while sand content is not sensitive to management, it does limit the ability of the soil to store or sequester carbon. In Figure 5, we show the relationship between sand content and SOC in the two LDSF sites included in the study. The trend of decreasing SOC with increasing sand content in these data is well established and has been reported in other studies using the LDSF from Tanzania (Winowiecki et al., 2016). This relationship is related to factors such as

the surface area of soil mineral particles, which decreases with increasing sand content leaving less area that SOC can be absorbed onto. Acknowledging this influence on SOC and other key properties is important for understanding restoration potential in terms of soil health as well as climate change mitigation potential.

The boxplots in Figure 6 show both predicted $\delta^{13}C$ and SOC across vegetation structure classes in the two LDSF sites. Generally, we found the lowest SOC contents and also higher $\delta^{13}C$ values in cropland, indicating SOC derived from C4 vegetation such as maize (*Zea mais*). In contrast, in areas where SOC is derived from vegetation with a C3 photosynthetic pathway, such as woodlands and shrublands, we found higher SOC values. These results indicate an opportunity to increase SOC through management

practices. This is especially apparent when assessing the effect of soil erosion on SOC. Soil erosion prevalence was more prominent in woodland, shrubland and grassland LDSF plots in the two sites as compared to cropland plots. This might indicate that farmers are already managing for erosion, which is an essential first step in building soil health, including maintaining and building SOC. Seventy-six percent of cropland plots were scored as not having severe erosion, with 24% having severe erosion.

Despite SOC variation in both categories (severely eroded and not severely eroded), there was no statistical difference in SOC content between these two. This finding differs from other studies that found erosion to have a strong effect on SOC content and stocks (Vågen and Winowiecki, 2013; Winowiecki et al., 2016). Both Nyagatare and Kayonza sites had low overall soil pH and exchangeable bases. However, these data are in line with what Vågen et al. (2016) reported using data from 114 LDSF

sites across sub-Saharan Africa (SSA), e.g., their results showed an overall mean topsoil OC of 22 g kg-1, a mean pH value of 6.1 and a mean sum of bases of 15 cmol$_c$kg$^{-1}$. Since very few plots were sampled

under naturally vegetated, undisturbed sites, our analysis is limited in terms of extending this into semi-natural systems. This was also reflected in the C3-C4 signal in the $\delta^{13}C$ data, which mostly indicated mixed C3-C4 systems. This highlights the need to use multiple indicators to understand drivers of SOC
dynamics, including interactions between plant communities, management, and inherent soil properties.

Field-saturated hydraulic conductivity (Kfs) was highly variable in the two study sites, as shown in Figure 7, with Nyagatare having slightly lower Kfs rates than Kayonza. SOC positively influenced Kfs, which is in agreement with previous findings highlighting the importance of soil organic matter for soil
aggregation and water infiltration (Franzluebbers, 2002). Our results indicate that sand content influences Kfs negatively, which is counterintuitive, as coarse-textured soils tend to have higher Kfs compared to more fine-grained soils (Hillel, 1980; García Gutiérrez et al. 2017). However, soil hydraulic properties of soils with finer textures have been shown to be less dependent on particle size distribution (García Gutiérrez et al. 2017), which could partially explain our results considering that
sand content in the plots where infiltration was measured was relatively low. It is also likely that the negative relationship between sand content and Kfs we have found reflects the positive effect of SOC on Kfs, as SOC and sand content had a strong negative relationship. On the other hand, soil pH and the presence of erosion had a negative effect on Kfs. Erosion and land degradation often lead to reduced soil infiltration capacity due to a decline in SOC and subsequent deterioration of soil structure (Valentin
& Bresson,1997), which in turn can result in increased infiltration-excess overland flow and further erosion (Blake et al., 2018). Our findings indicate the complexity in determining hydrologic controls across landscapes, which is something that will need to be studied in more detail in the future. Maintaining and promoting soil hydrological functioning is critical for food and water security and to build resilience to climate change (Bossio et al., 2010,Falkenmark & Rockström 2008, Cole et al. 2008),
but this is often overlooked in the discussions around restoration. Findings from this study highlight the importance of human-induced drivers on Kfs and, therefore, the potential to actively maintain and restore soil hydrological functioning.

Findings from this study demonstrate that by applying a consistent indicator framework such as the LDSF, which combines systematic field measurements with innovative laboratory methods, advanced data analytics and remote sensing, we are able to conduct spatial assessments of SOC, erosion and other land health indicators with high levels of accuracy. Such assessments and maps have applications not only for targeting land restoration interventions, but also for tracking changes in soil and land health
over time. For example, by mapping SOC at 30-m resolution, we can pick up spatial patterns related to both land management and inherent soil properties to identify both drivers of land degradation and land restoration potential, including SOC sequestration.

In a case study from the Lake Kivu area of Rwanda (Akayezu et al., 2020) showed the utility of erosion
hotspot mapping for spatial targeting of soil and water conservation measures. The results of the study presented here can be used in a similar manner to identify hotspots within the study area where erosion is occurring (Figure 8). These hotspots can in turn be combined with spatial assessments of SOC

(Figure 10) to more effectively target areas for land restoration, particularly where there is high erosion prevalence and low SOC. This is critically important, particularly if we consider the often high economic costs of restoring degraded land (Quillérou and Thomas, 2012) and the importance of land restoration for achieving the Sustainable Development Goals (Herrick et al., 2019). Furthermore, by combining spatially explicit indicators of land and soil health, spatial prioritization of restoration potential based on biophysical characteristics can enable decision making (Winowiecki et al., 2018).

Land degradation and restoration of degraded lands are complex processes that cannot be addressed effectively without considering multiple factors determining soil and land health. In this study we have assessed multiple indicators that can be readily quantified, and are widely accepted as important in determining soil and land health. Further, we used a sampling design that allowed us to measure these indicators consistently. This is critical for the design of interventions that target multiple aspects of land restoration, including soil erosion, species diversity and SOC. Specifically, this study identified low tree diversity and high occurrence of exotic timber species, highlighting an opportunity to explore the inclusion of indigenous tree species in both landscapes. In addition, the maps of soil erosion can be used to spatially target soil water conservation measures as well as set a baseline for tracking degradation over time. We argue that assessing these multiple indicators within a robust yet rapid sampling design will improve the effectiveness of restoration interventions as well as provide a baseline for tracking progress overtime.

## 5. Conclusions

We demonstrate the utility of systematic, multi-scale assessments of soil and land health across landscapes to target and monitor ecosystem restoration interventions, including the importance of understanding the interactions between indicators. By using a robust set of soil and land health indicators that are consistently sampled and characterized, we are able to provide analysis and spatial assessments at scales relevant to smallholder farmers. In the current study, we illustrate the approach with examples for SOC and erosion, although additional indicators may be included to address the complexity of land degradation and tailor land restoration interventions that consider interactions of multiple indicators in a spatially explicit way. We also demonstrate the importance of understanding both inherent and human-induced drivers of indicators such as SOC, which is critical for landscape restoration. We highlight the link between SOC, erosion, and hydrologic function. Using these data, we suggest land managers implement restoration options that reduce erosion, increase soil organic carbon and soil infiltration capacity, and increase aboveground biodiversity. Doing so has the potential to reach multiple goals, including food and nutrition security, climate change mitigation and adaptation, and biodiversity. We argue that there is an urgent need for systematic assessments of SOC, as well as aboveground biodiversity (e.g., tree diversity), combined with hydrologic properties and other indicators of land degradation such as soil erosion to effectively target interventions across landscapes. This will not only ensure that appropriate interventions for land restoration are implemented, but also provide the evidence base to assess their effectiveness.

Rwanda is one of the most progressive countries in the region in terms of acknowledging the importance of landscape restoration for sustainable livelihoods. The country has set ambitious targets over the next decade, aiming to restore more than 76% of its land area. Given the importance of the agricultural sector in the country and widespread land degradation due to a combination of deforestation and unsustainable agricultural practices, there is a need for evidence to support the targeting of land restoration efforts, as well as for tracking of the effectiveness of such interventions over time. By combining systematic field-based surveys with advances in soil spectroscopy and earth observation data, we can model and map SOC concentrations with high accuracy, allowing us to identify areas for restoration and track interventions over time.


## 6. Data Availability

All LDSF data are posted here:
https://data.worldagroforestry.org/dataverse/icraf_soils


## 7. Sample Availability

All soil samples are logged and barcoded at the ICRAF Soil-Plant Spectral Diagnostics Laboratory at World Agroforestry (ICRAF) in Nairobi, Kenya.


## 8. Team List

See acknowledgements and author list.

## 9. Author Contribution

LW and TGV co-led the conceptualization, writing and analysis. AM and AlexM led the coordination of the field work and contributed to the writing. PM contributed to collection of field data and interpretation. EBN and AlexM contributed to interpretation of the results. SC contributed to the contextualization. ATB contributed to the analysis, in particular to the infiltration analysis and modeling as well as the writing. LW prepared the manuscript with contributions from all co-authors.


## 10. Competing Interest

"The authors declare that they have no conflict of interest."

## 11. Disclaimer

The views expressed in this paper do not necessarily reflect the views of the donors who funded this work.

### 11.1.      Special Issue Statement

This article is part of the Special Issue: *Tropical Biogeochemistry of soils in the Congo Basin and the African Great Lakes region.*

### 11.2.      Acknowledgements

The authors would like to thank the partners who participated in the LDSF field surveys, including World Vision-Rwanda staff including: Jeremie Harerima, Donatien Niyibigira, Patrick Rugema, Thomas Habanabakize, Lambert Bucyana, Gilberrt Abakundanye, Augustin Tuyiturriki. Rwanda Agriculture Body, including Lambert Musengimana, Thomas Gakwavu, and Mukobwa Bijou, as well as Minani Vedaste from the Forestry Centre and John Thiongo Maina. We also acknowledge the staff of the ICRAF Soil-Plant Diagnostic Laboratory for the logging and organizing the soil samples as well as analysis of MIR Spectra. These project activities were supported through the funding of the European Union within the project dubbed, "Regreening Africa", grant number DCI-ENV/2017/387-627. We also acknowledge funding from the Swedish Research council Formas, grant number 2017-00430. Finally, this research was funded in part by the CGIAR Research Programme (CRP) Forests, Trees and Agroforestry (FTA) and the CGIAR Research Programme (CRP) on Water, Land and Ecosystems.

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

**Table 1:Prediction model performance metrics for the prediction of soil properties from MIR spectroscopy included in the study.**

| Soil property | $R^2$ | | RMSEP | |
|---|---|---|---|---|
| | Training | Testing | Training | Testing |
| SOC | 0.99 | 0.92 | 1.3 | 3.3 |
| d13C | 0.97 | 0.72 | 0.8 | 1.8 |
| pH | 0.97 | 0.84 | 0.2 | 0.4 |
| Sum of exchangeable bases | 0.96 | 0.84 | 3.9 | 8.2 |
| Sand | 0.98 | 0.84 | 3.1 | 8.9 |
| Clay | 0.98 | 0.82 | 3.5 | 10.1 |


**Table 2:  Soil properties for top and sub soil samples at the two LDSF sites (SD = standard deviation, ExBases is exchangeable bases) .**

| Site | Depth | N | Mean SOC | SD SOC | Mean d13C | SD dC13 | Mean pH | SD pH | Mean ExBaes | SD ExBases | Mean Sand | SD Sand | Mean Clay | SD Clay |
|---|---|---|---|---|---|---|---|---|---|---|---|---|---|---|
| | cm | | g kg$^{-1}$ | | ‰ | | | | cmol$_c$ kg$^{-1}$ | | % | | | |
| Kayonza | 0-20 | 151 | 20.9 | 8.83 | -18.9 | 1.15 | 5.65 | 0.68 | 10.3 | 8.69 | 19.8 | 9.29 | 58.4 | 11.5 |
| | 20-50 | 136 | 16.9 | 7.96 | -18.4 | 1.26 | 5.65 | 0.65 | 10.6 | 9.10 | 19.4 | 9.27 | 60.6 | 11.4 |
| Nyagatare | 0-20 | 149 | 17.3 | 6.07 | -19.2 | 0.92 | 5.89 | 0.54 | 8.74 | 4.80 | 30.0 | 10.2 | 44.5 | 10.5 |
| | 20-50 | 145 | 13.3 | 5.49 | -18.7 | 0.97 | 5.88 | 0.55 | 8.44 | 5.77 | 30.0 | 10.5 | 45.8 | 11.4 |


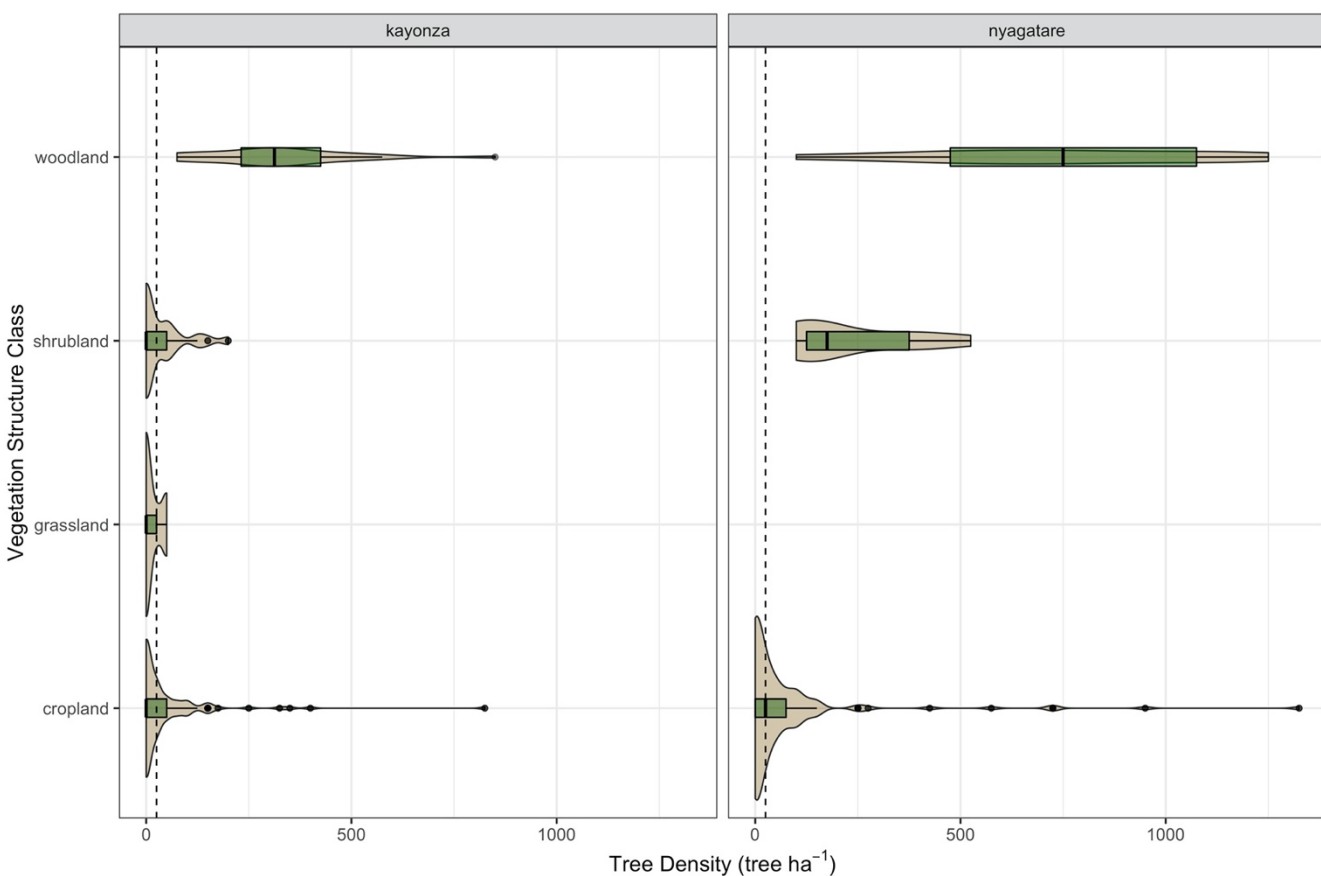

Figure 1: Violin plots showing the variation in tree densities across the vegetation classes at Kayonza and Nygatare LDSF sites, Rwanda. The dotted line is the overall median (25 tree ha$^{-1}$).


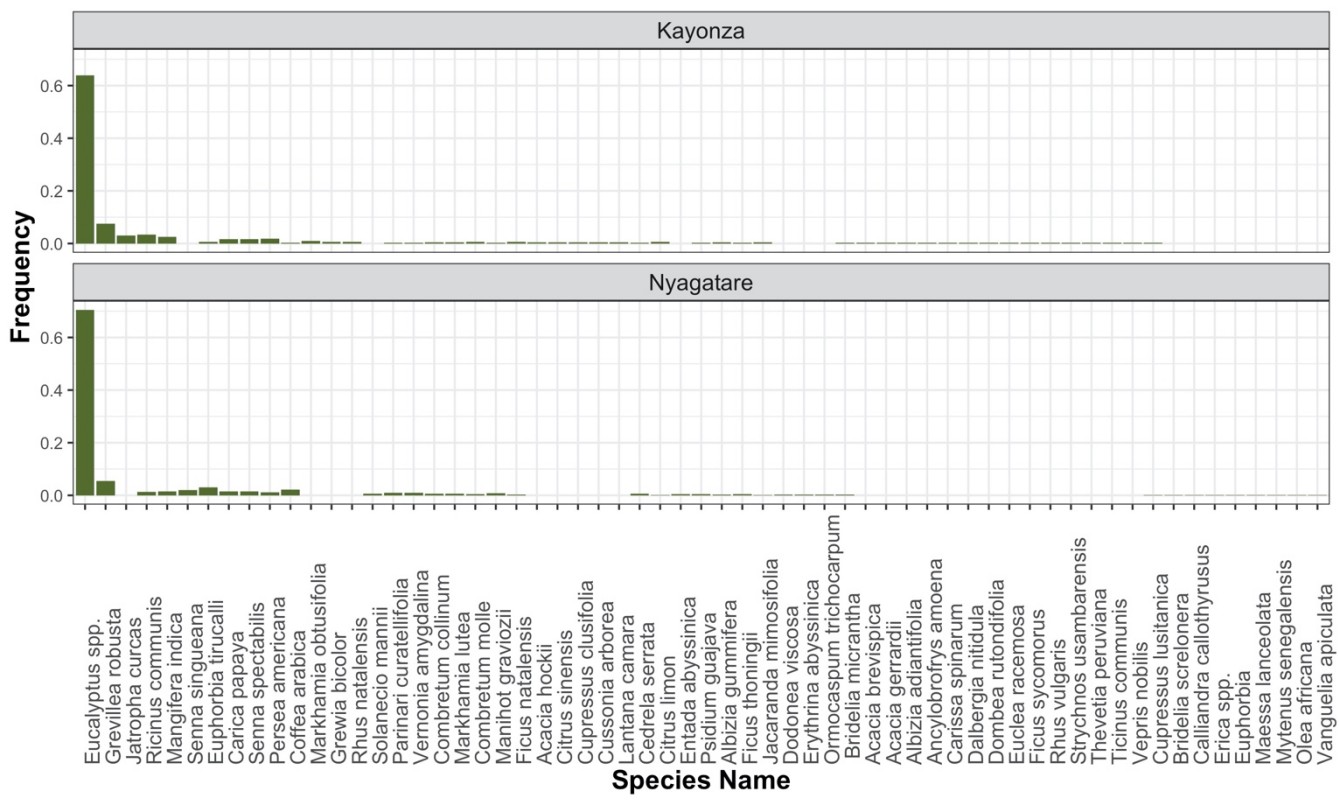

**Figure 2: Tree species across Kayonza and Nygatare LDSF sites, Rwanda. Sixty-two different species were recorded, with low occurrence of most species and few indigenous tree species.**


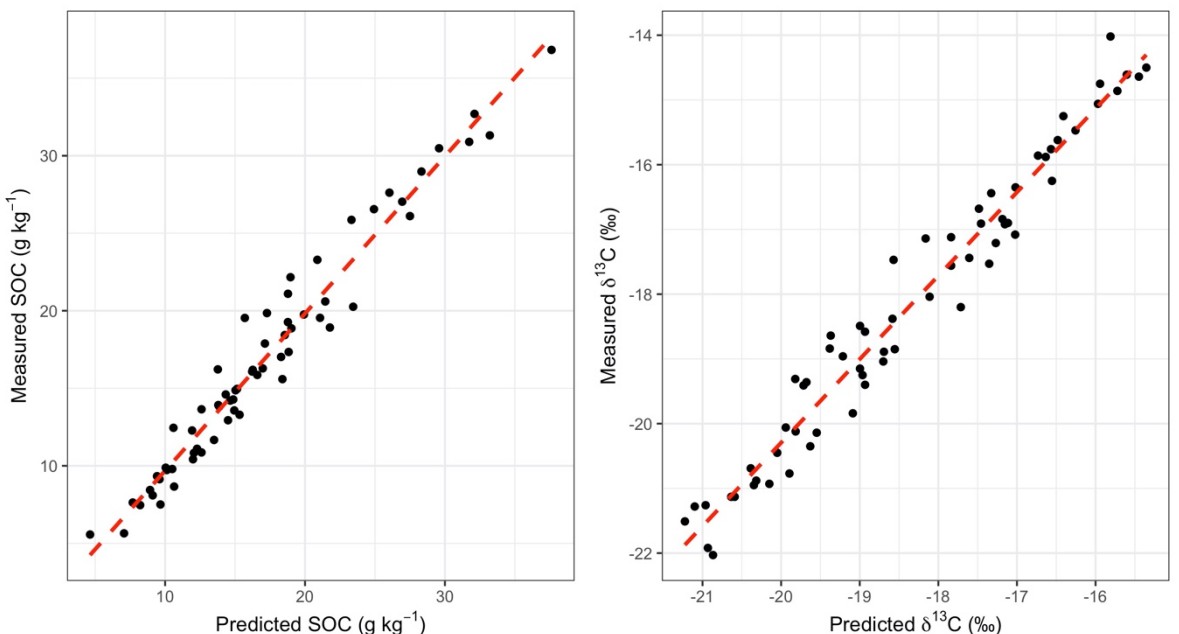

**Figure 3. Predicted vs measured SOC and δ¹³C based on MIR spectra for Kayonza and Nygatare LDSF sites, Rwanda.**


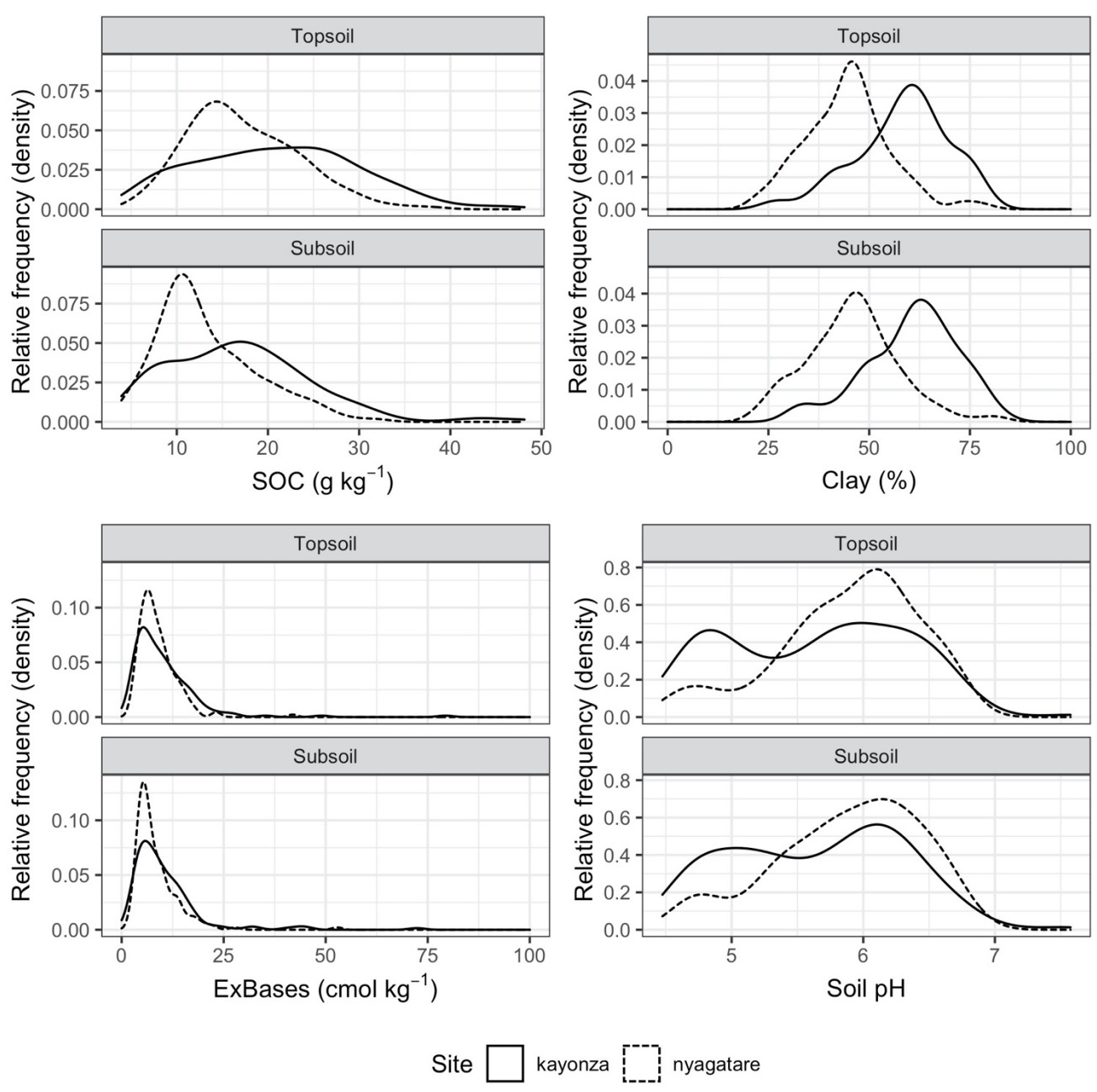

**Figure 4: Density plots of soil organic carbon (SOC), clay, exchangeable bases (ExBases), and pH for the top- and sub-soil samples at Kayonza and Nygatare LDSF sites, Rwanda.**


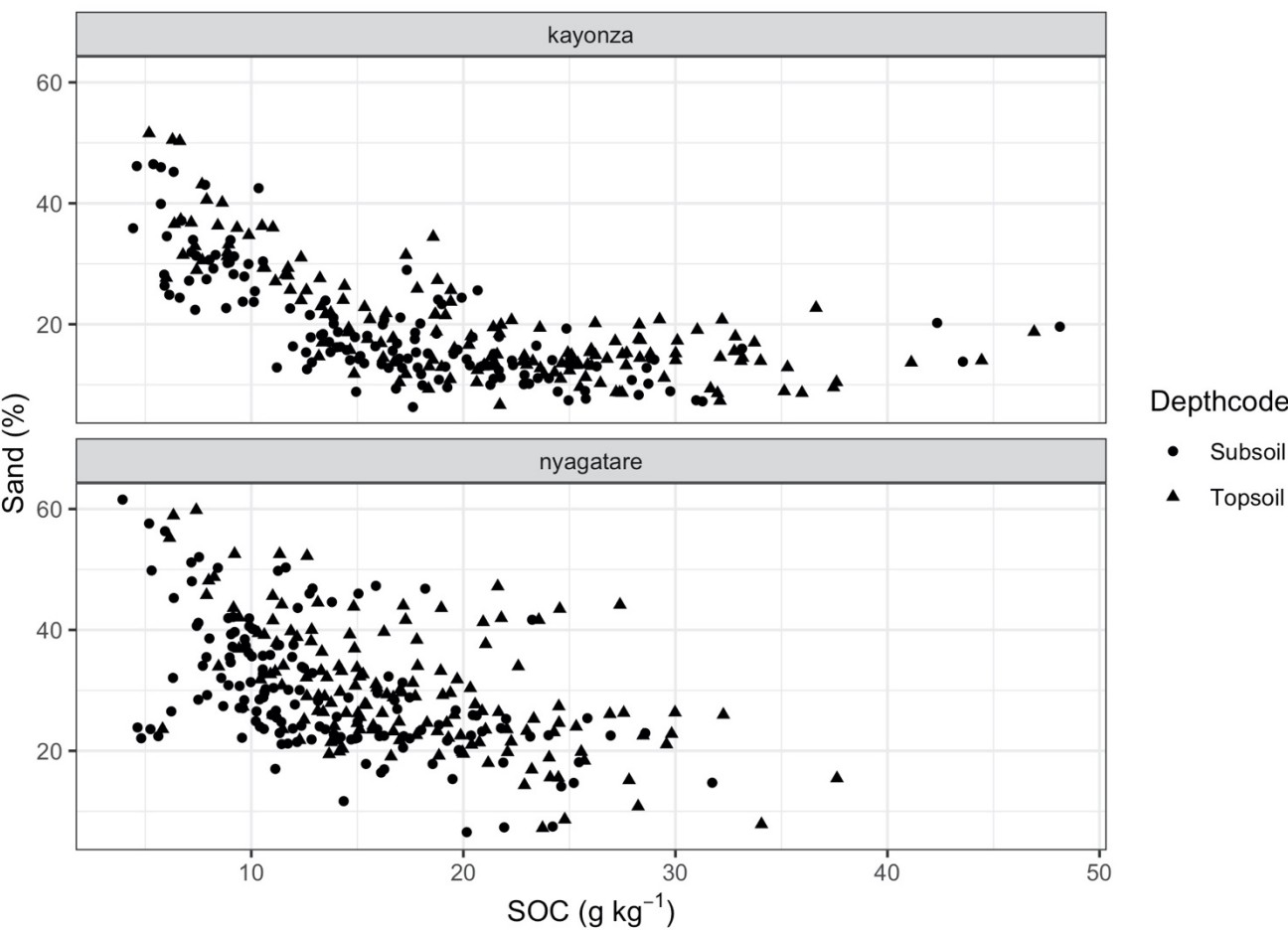

**Figure 5: Relationship between sand content and soil organic carbon (SOC) for both top- and sub-soil samples at Kayonza and Nyagatare LDSF sites, Rwanda.**




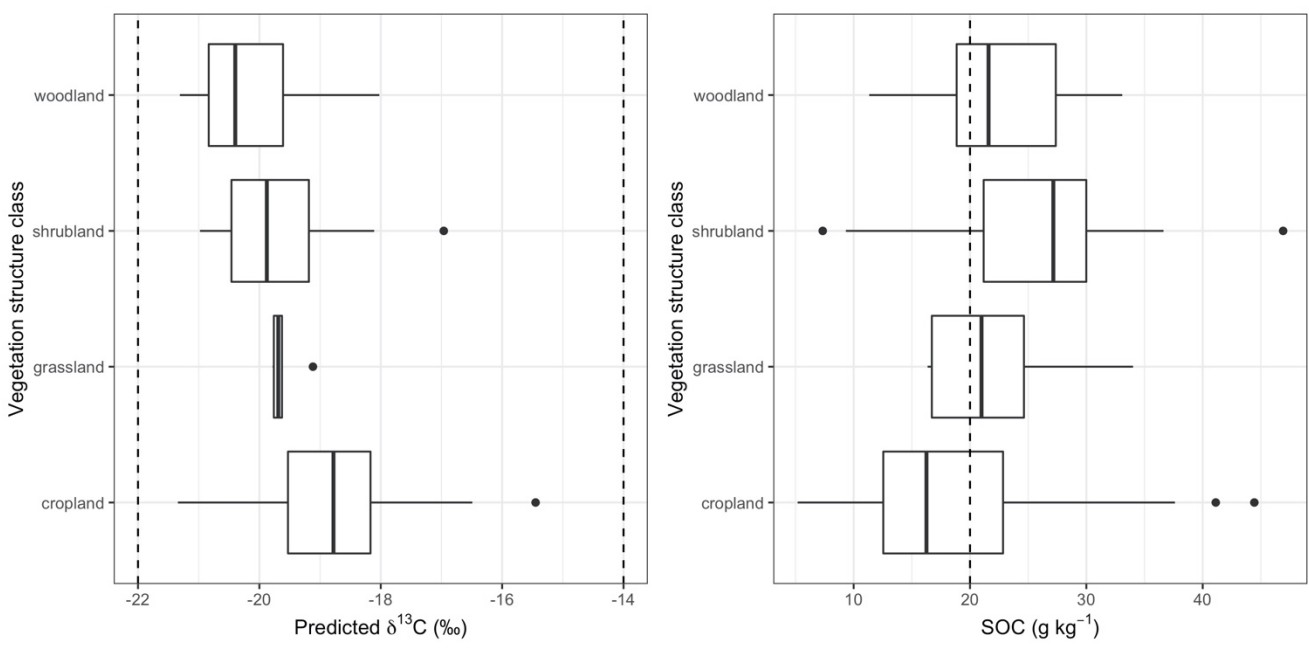


**Figure 6: Boxplots of δ¹³C values and soil organic carbon (SOC) content in topsoil for each vegetation structure class at Kayonza and Nyagatare LDSF sites, Rwanda. Dotted vertical lines at -22 and -14 ‰ δ¹³C indicate the C3 and C4 dominated systems, respectively. The dotted line at 20 g kg⁻¹ SOC is to indicate a threshold for agricultural productivity in humid areas.**


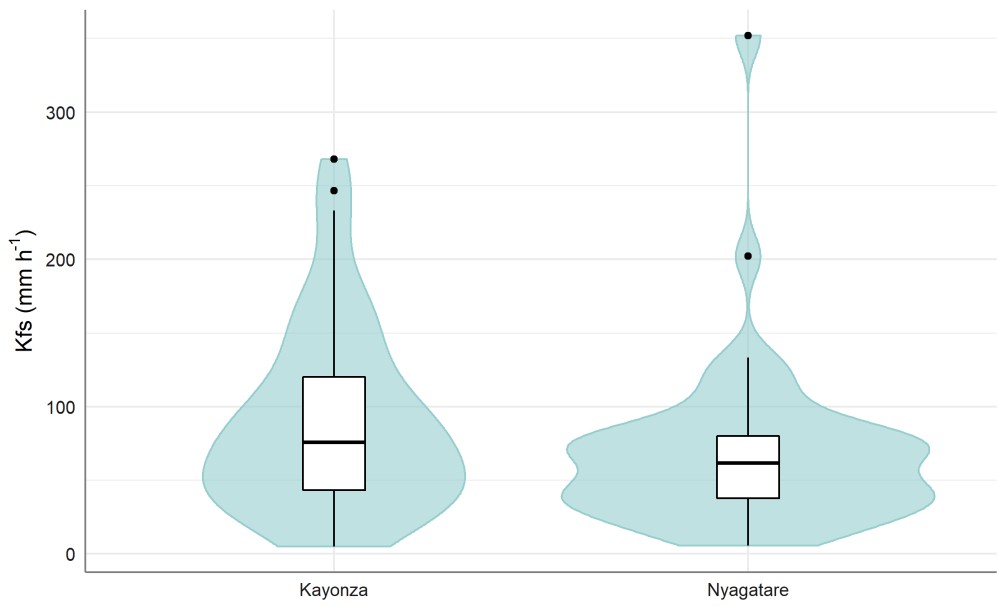


**Figure 7: Box and violin plots of field-saturated hydraulic conductivity (Kfs) for Kayonza and Nyagatare LDSF sites, Rwanda. The three horizontal lines in the box plot show the lower quartile, the median, and the upper quartile. Whiskers extend to the outer-most data point that falls within 1.5 box lengths. The violin plots show the distribution of the Kfs data.**


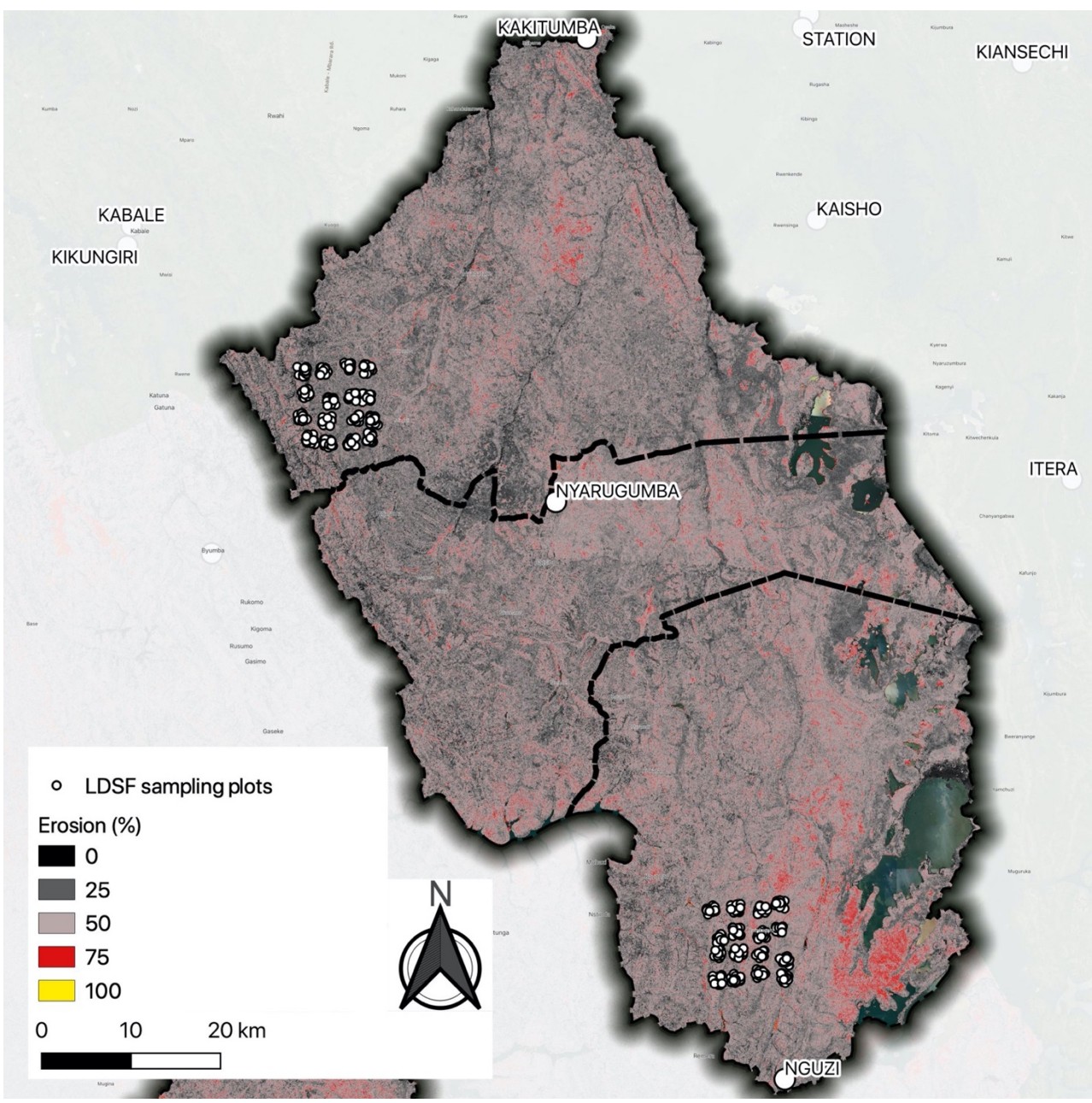


**Figure 8: Map of soil erosion prevalence (%) across Nyagatare, Gatsibo and Kayonza districts (Eastern province, Rwanda) predicted based on Landsat 8 satellite imagery and field data from the LDSF plots. The two LDSF sites are also shown on the map (Nyagatare in the north and Kayonza in the south), with the sampling plots shown as white circles.**

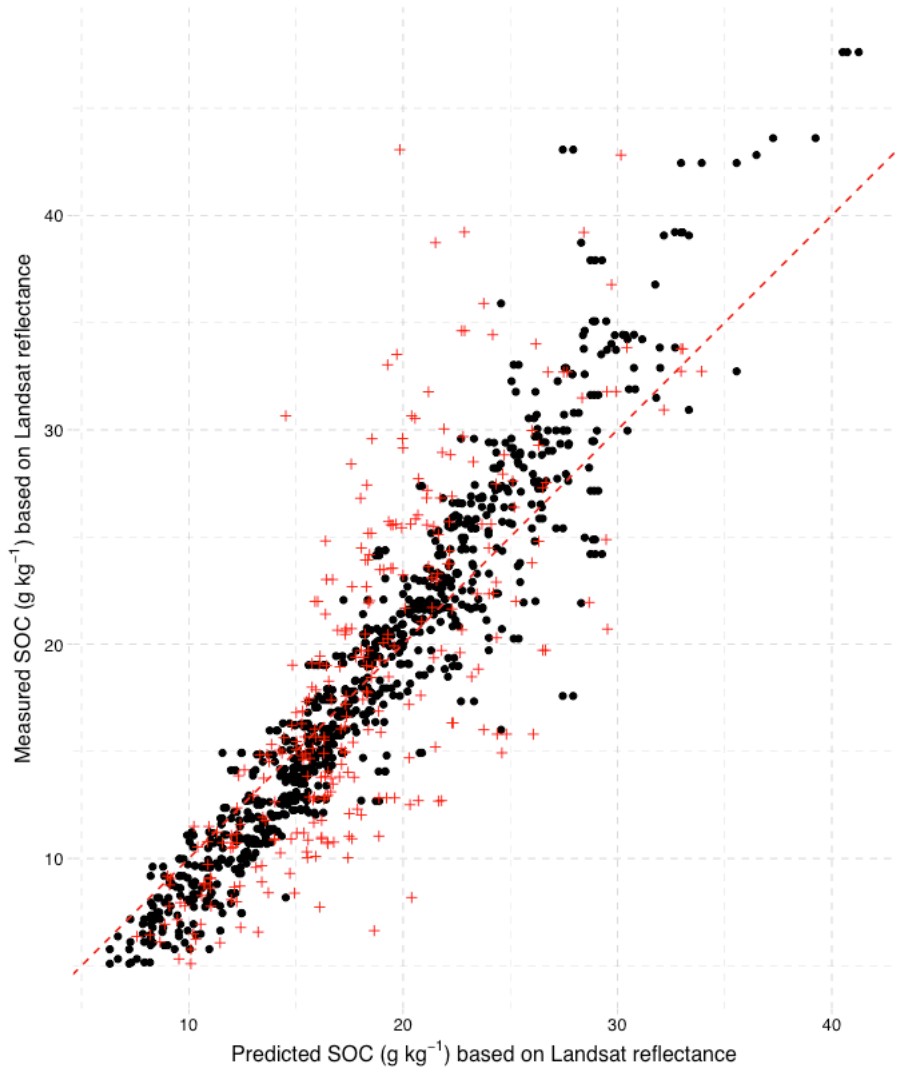


**Figure 9. Predicted vs measured SOC based on predictions made from Landsat 8 reflectance for Kayonza and Nyagatare LDSF sites, Rwanda. The black dots are training data, while the red crosses show independent validation results.**

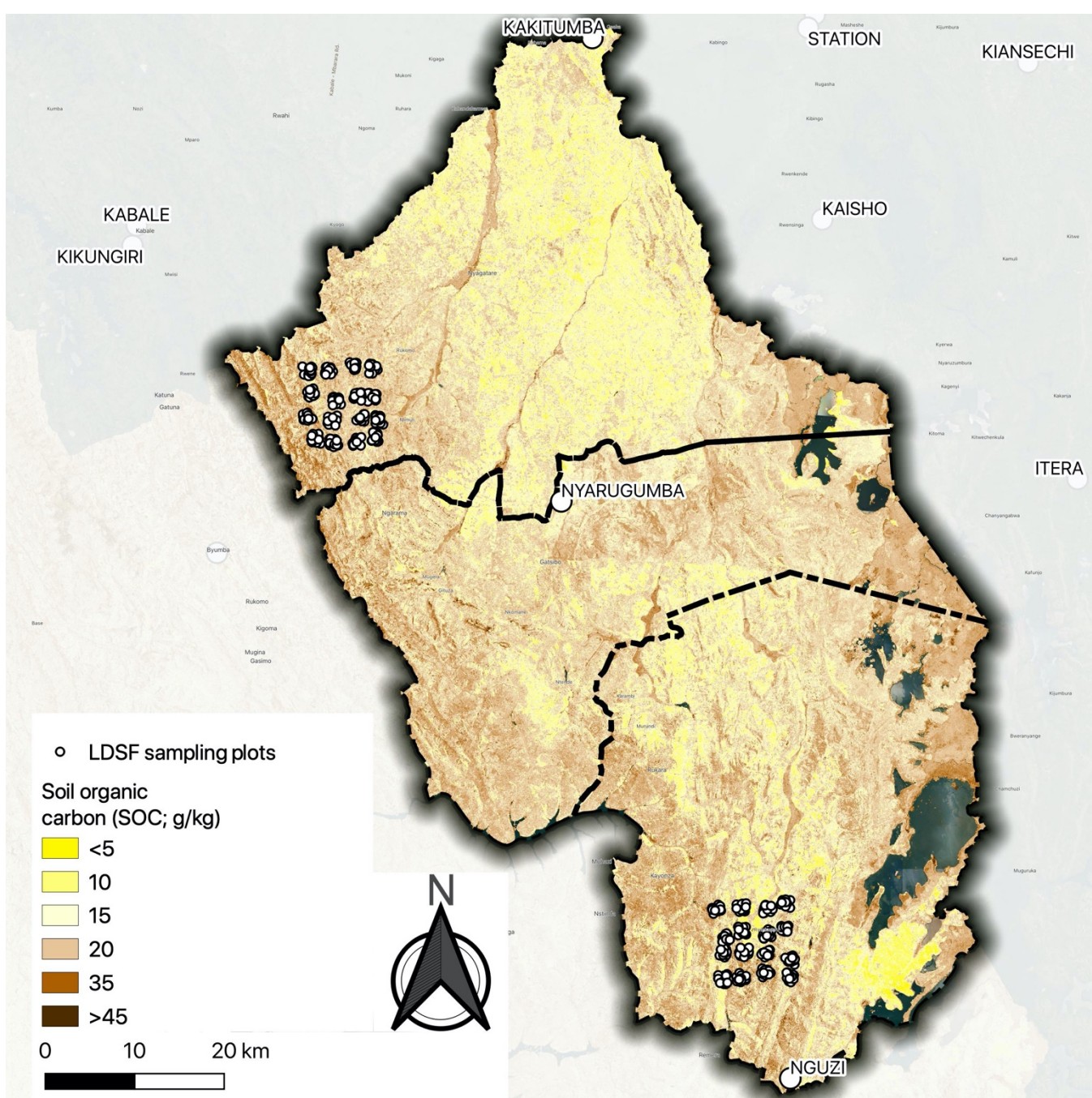

**Figure 10. Map of soil organic carbon (SOC) across Nyagatare, Gatsibo and Kayonza districts (Eastern province, Rwanda) predicted based on Landsat 8 satellite imagery and soil data from the LDSF plots. The two sites are also shown on the map (Nyagatare in the north and Kayonza in the south), with the sampling plots shown as white circles.**
