# Peer review of "Assessing soil and land health across two landscapes in eastern Rwanda to inform restoration activities"

_SOIL, 2020_

## Referee Comment (RC1) · Anonymous Referee #1 · 6 Jan 2021

The authors of this manuscript compiled an interesting dataset to assess "biogeochemical and human-induced drivers of soil organic carbon" (SOC) which might be used to evaluate soil restoration activities in Rwanda. Two districts of Rwanda were sampled in a stratified approach and about 150 samples were taken from topsoils and subsoils of each of the two districts. Various soil and site properties were determined and created maps of SOC contents and extent of soil erosion could be valuable tools to monitor future changes in soil restoration. Despite of this valuable dataset I have concerns to recommend publication of this manuscript. The most important criticism I have is a mismatch between the title, the introduction and the results / discussion of the manuscript. The authors wanted "to understand the extent of land degradation across two key action districts in Rwanda". The introduction of the paper is written to meet this motivation including to introduce the most important threat – soil erosion. However, the authors did not introduce the various indicators of soil health and land degradation including interactions between them. The second specific objective ("Understand the drivers of SOC dynamics") was not introduced at all although the title of the study indicates the potential importance of such drivers. In the section "Results" the authors presented the obtained data determined in the laboratory and in the field in a quite broad and general way which makes it very difficult to relate them to the extent of soil degradation and potential restoration activities. Furthermore, biogeochemical drivers of SOC were only discussed in a very superficial way without going into details of the processes and the existing literature. Therefore, the manuscript needs a complete revision / rewriting including a better focus of the paper, e.g. differences in the extent of soil degradation (including possible reasons for the differences) and potential consequences for restoration measures by using the indicators the authors determined. Data might be exploited to find interesting relationships between the determined indicators and to better understand the processes behind them. The controls of soil organic carbon should be an important focus of this data evaluation and interpretation.

In addition to this major criticism, I have further comments, which might be used to rewrite the paper: Title: Please find a title which really describes the main content / main message of your paper

Abstract: In the abstract you gave some interesting data, however, you did not relate the data to differences in soil degradation and their potential to be used for soil restoration measures. I miss a conclusion related to the main objective / title of the paper, i.e. relationships between drivers of soil C and restoration activities. Please use the appropriate format to report 13C ratios

Introduction: Please use the introduction to prepare the main objectives of your study. If you want to keep the focus as intended by the title, the more general parts of the introduction should be shortened substantially.

Methods: I miss details about the soils in your two districts, e.g. soil types, parent material. More details about soil sampling (auger?) and scoring as well as classification of soil erosion are necessary. Please add details about the instruments (EA, IRMS) and the location of the laboratory. The method to analyse soil texture was not described. It is not clear what kind of "full library" you used for calibration and prediction of MIR data (just from your study or more). The approach for soil mapping has to be described as well. This is a central part of the manuscript. It is not sufficient to give just a reference. Although the authors did some statistics, there is no description of the statistical approach. Statistics needs to be incorporated into the results section as well. It is not clear why the authors used both medians and mean values.

Results Often, median or average values of all land uses were given (please see my earlier comment regarding the use of these two parameters). I would prefer a presentation of the differences between the different land use systems as done in Figs. 1, 6. That would allow a better comparison between the two areas and with data from the literature. What is the reason for the higher tree density in one of the districts (120 vs 68 trees per ha)? I do not understand the database for a similar median of 25 trees per ha in the two districts. I would not describe the SOC contents of your soils as low – particularly SOC contents of your subsoils are quite high. Please distinguish between different land uses. You mentioned a high accuracy of your maps but what is the base of this statement? What does a R2=0.82 indicates? Is that the R2 between prediction of the model and measurements done in the framework of this study? Figure 5: change x and y axes Figure 9: database not clear: How was the prediction done and were the predicted values compared with measured values of the study?

Discussion This is a very weak section of the paper. It is not a discussion of your results related to the objectives as drivers of SOC and how they are related to restoration. You might explain and discuss differences in your measured indicators and what do these differences mean for soil health and restoration. Furthermore, you have to discuss these results based on the international literature. You found differences in topsoil

SOC contents between your districts. Discuss possible reasons for that. You might discuss potential effects of soil texture, pH and exchangeable bases on SOC. I would recommend to differentiate between topsoils and subsoils (also in Fig. 5). On the other hand, I would try to use the data of both districts in one regression. Do the effects of clay / sand on SOC depend on land use or on tree density? I would also add some discussion about reasons and consequences of the hotspots of soil erosion and low and high SOC contents. These are just some examples how the discussion might be rewritten.

In conclusion, I recommend major revisions of this manuscript, which means a complete rewriting.

---

## Referee Comment (RC2) · Anonymous Referee #2 · 20 Jan 2021

In this manuscript, the authors utilize the Land Degradation Surveillance Framework (LDSF,Vagen et al., 2016, Vagen and Winowiecki, 2020) to assess the variability of vegetation structure and diversity, soil carbon (total organic carbon and 13C) and other soil properties (texture, pH, exchangeable bases, and saturated hydraulic conductivity) in response to land use/vegetation type across two sites in eastern Rwanda (Nyagatare and Kayonza). The biophysical data from these LDSF sites was also used to generate spatial predictions of soil organic carbon and soil erosion prevalence across eastern Rwanda through the use of Landsat 8 imagery. This work is important and could be quite useful to managers and policy-makers, particularly in Rwanda, where restoration goals are uniquely ambitious. Therefore, the authors should be encouraged that this

manuscript is deserving of eventual publication. However, I have concerns with the manuscript in the current state that together amount to a significant revision of the manuscript and writing. My major comments are listed first followed by minor/editorial comments.

1. The title is compelling and the introduction nicely frames the importance of this work in the context of national and international policy goals. However, the results and discussion can be bolstered and better aligned with the title and objectives of the manuscript. The discussion is currently the weakest part of the manuscript and is poorly referenced. One suggestion for better aligning the discussion text with the title and objectives might be to create sections within the discussion section that align with the stated objectives of the study in the last paragraph of the introduction. This would provide focus for introducing more context to the data in this study. For example, the first objective is to assess soil and land health parameters across sites. Here perhaps the discussion could focus on why significant differences may or may not occur between the LDSF sites or vegetation classes investigated here (for example, what factors contribute to lower SOC in Nyagatare specifically? is one of the main drivers climate? or soil texture, geomorphology, etc...how does this relate to Kfs and what are the management implications of that?), and how the range of values compares to other landscapes in sub-Saharan Africa. The second objective is to understand the drivers of SOC dynamics. Given the dataset, the major drivers that could be assessed are geomorphic (slope - slope is mentioned twice, but never quantitatively explored as a driver - why?), climatic (rainfall?), erosion prevalence, land use, soil texture, and (maybe?) pH - although whether pH drives SOC accumulation (possible in some soil systems) or vice versa (SOC drives pH) may depend on the system. Which of these appears to be most important? What does that means for managers and policy-makers? The third objective is to develop hot-spot maps of soil erosion and soil organic carbon for targeting interventions - in the discussion more context regarding the major insights provided by these maps would be very helpful. How does the variability of these spatial predictions compare to other LDSF landscapes for example? Does that have any

impact on the importance of these spatial predictions for management?

2. I'm wondering if it might be possible to provide some tests of statistical significance between groups for the measured properties. This would help bolster the results, and allow the reader to better interpret the stated differences in means between sites and vegetation classes. For example, is topsoil SOC in Kayonza significantly higher than in Nyagatare? Or is it just numerically higher, given the variability. There should also be a portion of the methods that describes the statistical approaches utilized.

3. I do believe it is important not to ask readers to mine too many references when uncovering the methodology utilized. The LDSF methodology is well documented, but nevertheless there could be a more specific overview of soil sampling methods and geospatial techniques in the methods. These don't need to be pages worth of information, but a more full description of the methods employed here (even if they are published elsewhere) would be important for readers, particularly when it comes to cross-journal accessibility issues for international audiences. Yes readers can request documents from the author if necessary, but that places and undue burden on the reader.

4. What do the differences in soil properties between vegetation classes within sites tell us? Kayonza district, on average, appears to be wetter. Is there an interaction between climate, vegetation class, and SOC?

5. Several other things could be added to the methods: 1) it would be helpful to have a general description of the soils/soil parent materials across the region (and specifically within the sites), to include general soil types per FAO WRB system for international audience. 2) I'm wondering what the authors think about placing the vegetation structure and diversity section of the results (Section 3.1) in the methods? Given that much of this is background and that vegetation classes are utilized as potential drivers of OC in the interpretation of the data it seems like it may more appropriate there. I'm also not sure that Figure 2 is necessary as a plot given that after the first 5 or so most frequent species it is difficult to tell any differences between other species on the list. This might be more meaningful and useful as a list of species and frequencies (perhaps also broken up by land use?) in the supplementary material.

6. The MIR predictions perform very well. However, 13C performance is more likely due to correlations (what does a plot of 13C vs organic C look like?) than a reflection of MIR detecting isotopic differences in OC. Therefore, I'm wondering if the authors might use "estimated" before 13C just to indicate that these are estimated from MIR and no directly measured. I do understand that the other properties are also estimated in the same way, but I don't think it is as critical to make that distinction for the reader for those because MIR is much more likely to be giving more direct measures of the other soil properties.

MINOR COMMENTS: 7. The introduction could be re-arranged to improve readability. For example, paragraphs starting at lines 70 and 93 are both about soil erosion but separated by a paragraph on management strategies and agroforestry.

8. Ensure 13C notation and symbology is correct.

9. I do not believe that figures 4 and 8 are referenced in the text

10. Line 28 and Line 330: "compaction" is mentioned in the text here, but nowhere else in the manuscript. The way that it is mentioned in lines 28 and 330 appears to indicate that compaction was a measured variable and also that it had a direct impact on Kfs. However, I can't find reference to the compaction data or any correlation, plot or values for compaction in the reported numbers. Nor can I find a method for measuring compaction (cone penetrometer?). Compaction data should either be added to the manuscript or the word compaction should be removed from the text.

11. Line 39: Replace "In other words" with "Therefore"

12. Line 72 replace "execrated" with "exacerbated"

13. Line 84: insert "that" between "effects and "intensive"

14. Line 86: insert "to" between "access" and "labour"

15. Lines 89-90: replace "tree-based ecosystem" with "agroforestry"?

16. Line 124: replace "scale scale" with "spatial scale"

17. Line 147 and 155: deforestation rate percentages are mentioned. what are these rates describing: deforestation of total forested lands? more context to the meaning of the numbers would be good.

18. Line 151: there should be a reference at the end of this sentence.

19. Lines 176-177: more detail on the scoring of erosion prevalence would be extremely important. See comment 3, above.

20. Lines 210-220: information about laboratory soil particle size analysis for validation set should be provided here.

21. Lines 301-303. This would be an excellent sentence to include in a re-written discussion section.

22. Line 382: replace "religiously" with "systematically".

---

## Author Comment (AC1) · 1 Mar 2021

Thank you for the comments on the manuscript, which are very valuable.

Comment 1: We agree that the discussion can be expanded and strengthened, particularly with regards to aligning it with the title of the manuscript. We will also add additional references as suggested. We are expanding the discussion around site-specific effects as suggested, including a more in-depth analysis of interactions and statistical tests. This includes including a discussion around implications for management and how these data compare to other landscapes in SSA. We will further explore the main drivers of SOC, as suggested.

Comment 2: We will include additional statistical tests (in the methods and the results) to explore significance of key variables, including SOC.

Comment 3: We will also expand the section on the LDSF in order to reduce the effort required by readers to familiarize themselves with the methodology from other papers/sources.

Comment 4: Yes, we will better explore the effect of climate, great suggestion.

Comment 5: We will also expand on the methods section, providing more details as requested. We will look for available soil classification systems to include as site description. Since the vegetation classes are results of the field surveys, we will keep this in the results but perhaps include other references for the vegetation description in the site description within the methods. Thanks for the suggestion to turn Figure 2 into a table.

Comment 6: On MIR predictions and correlations between SOC and d13C. We will certainly explore this for this particular data set. However, it does not seem to be the case that SOC and d13C are strongly correlated, therefore we do think the prediction are detecting isotopic differences. We will add a figure for this and also include a more thorough discussion.

Thanks for the many minor comments and suggestions, which are very good, and also for noticing some missing cross-references, which we will correct.

---

## Author Comment (AC2) · 2 Mar 2021

Dear Reviewer #1,

Thank you for these comments. These were very helpful and will guide us in improving the paper. We have gone through each one of your valuable comments and agree that a comprehensive re-write and expansion will greatly improve the paper. We have addressed each one of your comments below and have taken action on the same in the updated version of the manuscript.

Yes, we will better integrate the indicators of soil and land health, beyond soil erosion.

[Figure]

This point is well taken. We are now working to better expand the section on drivers of SOC. This includes a better thread from the introduction through to the results and discussion. This includes an assessment of the controls of SOC.

We will include a more thorough and detailed analysis on the differences of the extent of soil degradation and consequences for restoration.

Regarding your comments to the specific sections:

Abstract: Yes, we will better highlight the major conclusions around drivers of SOC and its connection to restoration activities.

Yes, we will fix the format for reporting 13C ratios.

We agree, and will remove the mention of "low" SOC in the abstract.

Introduction: We will modify the focus and shorten the more general aspects and include more detail on indicators and SOC as suggested above.

Methods: We will include a more comprehensive site description, including soil types etc. We will expand the explanation of the LDSF sampling design, and specify the soil collection, e.g., soil augering. We did include the location of the laboratories, but can perhaps make this more clear. We will also expand the description of the MIR database, and how many samples were included in the calibration and validation datasets. We agree that this section is currently general and can be expanded. We will also expand the soil mapping approach used and the statistical analysis. These areas will be elaborated greatly.

Results: Agreed, we will stick to using just one measure, median, for example and not use both mean and median. We will further explore the differences between land uses, as suggested. The difference in tree densities was due to Eucalyptus plantations, but this was not explicitly mentioned in the text.

Yes, we will expand the soil mapping section, both in the methods and results, including

the data used in Figure 9. These sections will be greatly expanded.

Discussion: We are completely expanding the discussion section as per your suggestions. With more attention and focus on drivers of SOC and how they are related to restoration. We will also better discuss the differences in the measured indicators and the implications for soil health and restoration. We will also include more international references.

Thanks for the suggestion to combine the data and look at the effects of texture, pH and bases on SOC. We will expand on the interpretation of the hotspot map as well.

We thank you again for the thoughtful and thorough comments.

---

## Author Response (AR1)

**Topical Editor Decision: Reconsider after major revisions** (08 Mar 2021) by Sebastian Doetterl
Comments to the Author:
Dear authors,

Thank you for providing answers to the reviewer comments and thank you as well to the two reviewers for their helpful and extensive feedback. After evaluating both reviews and your response I recommend to go forward with a major revision for this manuscript which involves reworking the manuscript and in parts more analyses to provide further details to methods and the interpretation of your results.

In particular, I think it is important to address the comments of both reviewers considering the presentation and statistical analyses of your data (improving methods description and in parts revising the way the results have been presented); and to provide a substantially revised discussion, referenced with relevant international literature for framing and comparison of your findings to former research in other tropical regions.

While these comments involve extensive revision of the text, I do believe their implementation is straight forward since the reviewers gave very detailed advice. Additionally, I want to point out that the relevance of your work is high and I would much appreciated to see a revision of this manuscript. When providing the revision, please also provide a point-by-point response letter to all reviewer comments and a track-changes version of the manuscript for an easier tracing of the modifications made to the MS.

Non-public comments to the Author:
Dear authors, while I think the comments are easy to implement I have selected "Major Revision" in order to allow for idle time for the revision. Let me know if the timeline of the revision is too tight but I am confident this can be done in the given time with that decision.

Best wishes,
Sebastian Doetterl

**I.   *Comments to the Editor:**

Dear editor,
Thank you for the opportunity to revise the paper. As you can see from the track changes we have made major revisions, which we think greatly improved the focus of the paper. We took each of the reviewers' comments into account, and have provided a detailed account of how we addressed them  below. In general, we restructured the introduced to include a section on the role of soil and land health indicators for achieving the SGDs as well as climate change and restoration targets. We removed much of the background on Rwanda and moved some of this into the Methods section. We also removed the paragraph on the Regreening Africa project. We look forward to your feedback on the same. In the Methods, we expanded each section including LDSF description, statistical analysis, MIR predictions and soil mapping methodology. We expanded the results on the same. The discussion is completely rewritten.

We hope the reviewers will be satisfied with the complete revision, however we will be happy to address any further comments to finalize the publication.

Thanks again!
Leigh

**II.    Comments from Reviewer #1**

Thank you for these comments.

These were very helpful and have guided to improve the paper. We have gone through each one of your valuable comments and agree that a comprehensive re-write and expansion was needed to improve the paper.

We have addressed each one of your comments below and have taken action on the same in the updated version of the manuscript, kindly see track changes. Yes, we have better integrated the indicators of soil and land health, beyond soil erosion. And we have changed the title, replacing the word biogeochemical with inherent.

*R1 C1: "better focus on differences in the extent of soil degradation (including possible reasons for the differences) and potential consequences for restoration measures by using the indicators the authors determined"*
This point is well taken. We have expanded the section on drivers of SOC. This includes a better thread from the introduction through to the results and discussion. This includes an assessment of the controls of SOC.
We included a more thorough and detailed analysis on the differences of the extent of soil degradation and consequences for restoration .

Regarding your comments to the specific sections:

*R1. Abstract:*
Yes, we have now highlighted the major conclusions around drivers of SOC and its connection to restoration activities and specifically for ecosystem function.
Specifically, we highlight the relationship between SOC and sand and KfS and expand on the role of soil carbon on soil health, as well as the opportunity to increase SOC with land management. We also highlight the influence of vegetation structure on soil erosion. And the opportunity to include these data and information in the land restoration agenda.

Yes, we have fixed the format for reporting 13C ratios. -   $^{13}$C

We agree, and have removed the mention of "low" SOC in the abstract. Done

*Introduction:*
"Prepare the main objectives of your study to keep the focus as intended by the title"

We have completely restructured the introduction, we added a section on soil health and the link with landscape restoration.
We also included more detail and references on indicators and SOC as suggested.

We have shortened the more general aspects and most of the introduction to Rwanda.

*Methods:*
We have included a more comprehensive site description, including soil types, etc.
We expanded the explanation of the LDSF sampling design, specifically the randomization, the set-up of the subplots, the soil collection, e.g., soil augering and the soil erosion scoring.

We had included the location of the laboratories, but can perhaps make this more clear.

We also expanded the description of the MIR database, and how many samples were included in the calibration and validation datasets.

We also expanded the soil mapping approach used and the statistical analysis.

*Results:*
Agreed, we will stick to using just one measure, mean, for example and not use both mean and median.
We further explored the differences between land uses, as suggested. We ran statistical tests on the effects of vegetation structure on SOC: cropland~grassland<shrubland<woodland

The difference in tree densities was due to Eucalyptus plantations, but this was not explicitly mentioned in the text .Yes, we will expand the soil mapping section, both in the methods and results, including

These sections will be greatly expanded.

Discussion: We are completely expanding the discussion section as per your suggestions. With more attention and focus on drivers of SOC and how they are related to restoration. We will also better discuss the differences in the measured indicators and the implications for soil health and restoration. We will also include more international references.
Thanks for the suggestion to combine the data and look at the effects of texture, pHand bases on SOC. We will expand on the interpretation of the hotspot map as well.
We thank you again for the thoughtful and thorough comments.

Reviewer 2:
Thank you for the comments on the manuscript, which are very valuable.

Comment 1:
We have expanded and strengthened the discussion particularly with regards to aligning it with the title of the manuscript.
We will added additional references as suggested. Specifically we expanded the discussion around site-specific effects as suggested as well as the effects of vegetation structure. We also discussed implications for management and how these data compare to other landscapes in SSA.

Comment 2:
We have included additional statistical tests (in the methods and the results) to explore significance of key variables, including SOC. (SOC by site, by sand, and with vegetation structure). We ran anova test and yes SOC was statistically higher in Kayonza compared to Nyagatare (P<0.001) and higher in topsoil vs subsoil. We also used lmerTest library in R to asses differences in SOC by Vegetation Structure and the patterns were the same in the both sites, though the magnitude varied- cropland~grassland<shrubland<woodland.

Comment 3:
We expanded the section on the LDSF in order to reduce the effort required by readers to familiarize themselves with the methodology from other papers/sources. Including a better description of the scoring of erosion and soil sampling.

Comment 4:
Yes, we explored the effect of climate in the discussion.

Comment 5:
We expanded the methods section, providing more details as requested. We included available soil classification systems. Since the vegetation classes were results of the field surveys, we will keep this in the results but perhaps include other references for the vegetation description in the site description within the methods.

Regarding the suggestion to turn Figure 2 into a table. We have gone back and forth on this and have decided to keep it as it demonstrates a desperately low occurrence of indigenous species and a dominance of Eucalyptus across the sites. If the Editor insists we delete it, we will do so. Thanks.

Comment 6:
On MIR predictions and correlations between SOC and $^{13}$C. We have explored this for this particular data set. However, it does not seem to be the case that SOC and $^{13}$C are strongly correlated, therefore we do think the prediction are detecting isotopic differences.
I include a figure here for you to demonstrate the lack of correlation between $^{13}$C and SOC.

[Figure]

*Editorial comments:*

Thanks for the many minor comments and suggestions, we have actioned each edit.

Thanks also for noticing some missing cross-references, which we have corrected.

---

## Author Response (AR2)

Response to the Editor:

*Dear Editor,*

*We are glad to hear that the reviewers have noticed a significant improvement in the manuscript after our major revision. However, both reviewers have raised additional comments. We are grateful for these and for the opportunity to re-submit a revised version.*

*We are sorry about not having been able to satisfactorily address the major criticism reviewer 2 raised in the first review concerning a supposed mismatch between the title, the introduction and the results / discussion of the manuscript. We believe we have fully addressed these concerns in this version and the other comments from reviewers 2 and 1. Most notably, we have changed the title of the manuscript to remove the aspect of 'drivers'. We also expanded the explanation of additional indicators in the introduction (soil organic carbon, soil erosion, infiltration capacity, vegetation, etc). We have now added hypotheses, as requested. We have expanded the section on the LDSF as well. Finally the discussion on the utility of these indicators for assessing soil degradation has now been included in the discussion..*

*We are very grateful to the reviewers for the effort and time in reviewing this manuscript. Their comments and inputs have been very valuable for us and have helped improve the original manuscript's quality. The response to each of the individual comments is below.*

*Thank you,*

*Leigh Winowiecki*

I.    Comments and Response: Reviewer #1

R1 C1: I highly suggested that the authors re-read the manuscript carefully and correct minor issue with tense (i.e. "is" vs "was, etc.), grammatical errors, and typos, which are minor but scattered throughout the manuscript. (Editor: Agreed. Please go through the MS top to bottom before re-submitting and check for consistency in writing)

*Thank you for pointing this out. We have now gone through the manuscript in detail and edited it to ensure consistency in the use of verbal tenses. We have also corrected other minor grammatical mistakes and typos.*

II.    Comments and Responses: Reviewer #2

R2 C1: In the response letter, the authors did not really indicate how they have addressed my major criticism. (Editor: I noticed that the response letter from the last round was quite brief in

this regard. Please explain in your response letter in detail how you address these issues and where to find your changes in the document. Bring examples if changes were done throughout the manuscript and not just for a specific issue. Please also explain in the response letter more explicit why certain comments can be ignored / do not need to be addressed in the extent suggested by the reviewers.

*Our apologies for not adequately addressing these issues in the previous response letter. We have included a track changes document whichs shows where specific changes were made, as well as a letter to the editor with a general response on the changes and a detailed response to each comment outlining how each comment was addressed. We hope this provides sufficient details on how we explicitly addressed each of the reviewers' comments.*

*We have modified the title to remove the aspect of drivers and to more accurately reflect the content of the manuscript. The new suggested title is, "Assessing soil and land health across two landscapes in eastern Rwanda to inform land restoration activities". This is a direct response to the reviewer's comment that drivers were not addressed and that the title did not reflect the context of the manuscript. We hope the reviewer and editor are satisfied with this change.*

*In the abstract we have now added these two sentences to further clarify the objective of the paper., "*These data demonstrate the importance of assessing multiple biophysical properties in order to understand land degradation, including the spatial patterns of soil and land health indicators across the landscape. By understanding the dynamics of land degradation and interactions between biophysical indicators, we can better prioritize interventions that result in multiple benefits, as well as assess the impacts of restoration options.*"*

*We have expanded the introduction to include the explanation of the additional soil health indicators. In the original version, we focused on SOC, however as mentioned by the reviewer, we also included a number of biophysical variables to provide a more complete assessment. Specifically we included a description of the importance of considering multiple variables including, SOC, erosion prevalence, vegetation structure, tree density and species diversity, topsoil field-saturated hydraulic conductivity (a proxy for steady-state infiltration capacity), soil texture, pH and exchangeable bases. These additions were made to satisfy this comment from the reviewer "*However, the authors did not introduce the various indicators of soil health and land degradation including interactions between them.*"*

R2 C2: The authors did not study drivers of soil organic carbon although indicated in the title. (Editor: Agreed. The manuscript is rather on SOC and other soil parameters as an indicator for a specific soil status and soil health in general, but not focused so much on the aspects of drivers as suggested by the title)

*We understand the reviewer's concern that there was a mismatch between the title, the introduction, results and discussion section in the first version of the manuscript, and that perhaps the original title was not adequately reflecting the full scope of the study, as it was referring to drivers of SOC only. This concern has been addressed in detail in C1. Yet, we disagree with the reviewer's opinion that we did not study drivers of SOC, but we do agree that this was not the sole focus of the study. While our data clearly demonstrate the control of sand content on SOC, as well as the impact of vegetation structure on SOC, we have also included a number of other key indicators. Therefore, we have changed the title to "Assessing soil and land health across two landscapes in eastern Rwanda to inform restoration activities".*

R2 C3: I do not see that they expanded the section on drivers of SOC as indicated in the response letter. (Editor: Indeed, aspects of soil degradation and consequences for restoration were explained much better now. but I do agree that the section on drivers should give a broader picture still to guide the reader and put results into perspective).

*We have changed the title of the manuscript to more accurately reflect the content. Specifically, we have eliminated the aspect of drivers. Therefore have not elaborated on the drivers of SOC but instead elaborated on the various indicators to be used when assessing land degreadation and prioritizing restoration actions.*

R2 C4: In the introduction the authors discussed the shortcomings in the assessment of land degradation and restoration very correctly (starting in line 62) and they introduced SOC as a universal / key indicator. In the next paragraph (starting in line 75), the authors addressed different indicators without given any explanation what indicators they have in mind. That illustrates one of the shortcomings of this manuscript. In the introduction the authors imply that only one universal indicator is used, however, that is not the case giving the variety of different indicators they analyzed. (Editor: I agree that the introduction is only in parts aligned with what later is analyzed and presented. More information and guidance for the reader to arrive at what you describe the study will do in l. 117 and following is needed and this should include a brief description of the other indicators you plan to look at).

*We agree with the reviewer that the other indicators measured as part of the LDSF and included in this study need to be introduced here. We have now done this briefly in the introduction section in two different paragraphs (in one, we name all the indicators included in the study, and in the other, we give a brief intro about the most relevant ones ).*

*Lines 86-88 we added* "We argue that a coherent set of indicators collected using consistent measurement methods is needed to address the completely of ecosystem function."

*In addition to the mention of SOC and erosion we added the below from lines 93 to 118*

*"In addition exchangeable base cations provide a measure of available nutrients and soil pH provides a measurement of potential constraints such as acidity. Land cover and vegetation structure play a key role in terms of driving soil organic carbon dynamics in landscapes while also influencing land degradation processes such as soil erosion. Therefore, indicators such as tree density within various vegetation structure classes and overall tree diversity provide useful information for informing restoration interventions around reforestation (Di Sacco et al., 2020). The use of carbon isotopes provides further insights on vegetation shifts as $\delta^{13}C$ values in the soil reflect the photosynthetic pathway of the aboveground vegetation (Boutton et al. 1998). Soil infiltration capacity is another well-established indicator of soil health, in particular of the soil's physical status and its hydrological functioning (Allen et al., 2011). Soil infiltration capacity influences the recharge of soil and groundwater stores and the generation of surface runoff, with implications for erosion and flooding occurrence (Hillel, 1998)."*

R2 C5: I would expect from the manuscript an introduction / motivation of the used indicators as e.g. vegetation composition, tree density, species diversity, hydraulic conductivity, 13C. Unfortunately, that was not done. The authors should also decide whether SOC is an inherent soil property (e.g. line 123) or controlled by inherent drivers. (Editor: agreed)

*We have expanded the introduction on the use of various indicators. See above comment and*
lines 93 to 118
*In the introduction we also state, "However, ecosystems are complex and multiple biophysical and socio-economic factors need to be considered when targeting, planning, implementing and tracking restoration on the ground." We have also expanded on the various indicators used to both assess land degradation and prioritize land restoration activities.*
*Furthermore, we have cleared up an mis-understanding around SOC. For example by stating, "We assessed the relationship between inherent soil properties (such as texture) and SOC,…*

R2 C6: After a thorough introduction of the indicators, I would expect hypotheses related to these indicators and how they might be interrelated. (Editor: Agreed. The formulation of testable working hypotheses related to your objectives as listed in l.120-122 and the data you want to use for that would be important and quite helpful for a reader).

*We have now added these from lines 163*
*"Specific objectives of this study were to: 1) Assess soil and land health indicators across two landscapes; 2) Identify biophysical constraints; 3) Develop maps of soil erosion hotspots and variations in SOC for restoration interventions, based on the hypothesis that remote sensing (spectral) data can be used to predict erosion and SOC. We also assessed the relationship between inherent soil properties, such as texture, and SOC, the hypothesis being that factors such as sand content create constraint envelopes in terms of variations in SOC. Another hypothesis addressed in the study was related to whether there is a positive effect of SOC on*

*field-saturated hydraulic conductivity when we consider data from across diverse landscapes. We also assesses the influence of other soil properties on field-saturated hydraulic conductivity, in addition to human-induced processes such as soil erosion. Finally, we assessed the current status of vegetation structure across the landscape, in addition to tree density and tree species diversity, and conducted spatially-explicit assessments of SOC for eastern Rwanda."*

R2 C7: The general approach of the used framework (LDSF) has not been explained in detail although being decisive for the whole manuscript. I think it is much more than just the field sampling design. (Editor: I tend to disagree with this comment. LDSF has been described in the cited publications in great detail. However, the reviewer has a point. I think the fact that the manuscript does rely heavily on LDSF justifies to add a few extra lines on the motivation behind LDSF and how it relates to the study at hand).

*We greatly expanded the methods section, including a detailed description of the various measurements made within the LDSF in the first revision. In the current version we have added a paragraph in the methods from lines 320-325* **"The rationale behind the use of the LDSF in the current study was that it has been applied across a wide range of landscapes in the global tropics and has been shown to be robust in terms of assessing soil and land health in landscapes. It uses a standardized set of indicators that are consistently sampled and quantified, allowing for comparative studies between sites or landscapes. Also, the LDSF has been successfully applied in other studies for the mapping of indicators of soil and land health when used in combination with remote sensing satellite data (Vågen and Winowiecki, 2019, Vågen et al., 2013b)."**

R2 C8: In the discussion, the reader of the journal would expect an evaluation how useful these indicators are in assessing soil degradation. This part is too superficial. (Editor: Agreed. A thorough analysis of the usefulness but also the limitation and uncertainties related to the use of the suggested soil health indicators would strengthen the discussion greatly. Especially if it can be related to a case study such as the one presented here).

*We have added a discussion on the limitations and the usefulness on these indicators. Specifically from 676:*
*"Land degradation and restoration of degraded lands are complex processes that cannot be addressed effectively without considering multiple factors determining soil and land health. In this study we have assessed multiple indicators that can be readily quantified, and are widely accepted as important in determining soil and land health. Further, we used a sampling design that allowed us to measure these indicators consistently. This is critical for the design of interventions that target multiple aspects of land restoration, including soil erosion, species diversity and SOC. Specifically, this study identified low tree diversity and high occurrence of exotic timber species, highlighting an opportunity to explore the inclusion of indigenous tree species in both landscapes. In addition, maps of soil erosion will be used to target soil water*

*conservation measures to curb soil erosion. We argue that assessing these multiple indicators within a robust yet rapid sampling design will improve the effectiveness of restoration interventions as well as provide a baseline for tracking progress overtime."*

Additional note.
We have added the missing references as well.